# BOSE-NAS: Differentiable Neural Architecture Search with Bi-Level Optimization Stable Equilibrium

## Abstract

Recent research has significantly mitigated the performance collapse issue in Differentiable Architecture Search (DARTS) by either refining architecture parameters to better reflect the true strengths of operations or developing alternative metrics for evaluating operation significance. However, the actual role and impact of architecture parameters remain insufficiently explored, creating critical ambiguities in the search process. To address this gap, we conduct a rigorous theoretical analysis demonstrating that the change rate of architecture parameters reflects the sensitivity of the supernet's validation loss in architecture space, thereby influencing the derived architecture's performance by shaping supernet training dynamics. Building on these insights, we introduce the concept of a Stable Equilibrium State to capture the stability of the bi-level optimization process and propose the Equilibrium Influential ($E_{\mathcal{I}}$) metric to assess operation importance. By integrating these elements, we propose BOSE-NAS, a differentiable NAS approach that leverages the Stable Equilibrium State to identify the optimal state during the search process and derives the final architecture using the $E_{\mathcal{I}}$ metric. Extensive experiments across diverse datasets and search spaces demonstrate that BOSE-NAS achieves competitive test accuracy compared to state-of-the-art methods while significantly reducing search costs.

## 1 Introduction

Designing network architectures specifically tailored for specific tasks remains a formidable challenge. Recently, Neural Architecture Search (NAS) has become essential in automating the design of neural networks across various deep learning fields (Ren et al., 2021; Wu et al., 2021; Zhang et al., 2020; 2021a). However, early NAS methods were often computationally expensive (Zoph & Le, 2016; Real et al., 2019; Wang et al., 2020; Kandasamy et al., 2018), limiting their practical application. To achieve greater efficiency, recent advancements have adopted the one-shot paradigm, also known as weight-sharing (Pham et al., 2018; Bender et al., 2018; Liu et al., 2018b; Guo et al., 2020; Xie et al., 2019; Cai et al., 2018). A significant development within this paradigm is DARTS (Liu et al., 2018b), which integrates a continuous mixture of architectures transforming the architecture search problem into a differentiable task of learning architecture parameters then selects the operations corresponding to the largest parameter values at the end of the training phase to construct the final architecture. Despite its efficiency, DARTS has faced several challenges related to performance degradation issue (Liang et al., 2019; Wang et al., 2021; Zela et al., 2019; Zhang et al., 2022; Xue et al., 2022). Numerous studies (Ye et al., 2022; Chu et al., 2020b;a) have demonstrated that architecture parameters are negatively affected during supernet training, leading to various proposals for controlling or adjusting these parameters, which are fundamental to its selection rule, to better reflect the true strength of operations. However, recent literature suggests that the limitations in DARTS primarily stem from the inability of architecture parameters to accurately reflect the true strength of the operations, prompting the introduction of alternative metrics (Wang et al., 2021; Xiao et al., 2022; He et al., 2024). While these contributions have substantially alleviated the performance collapse issue inherent in DARTS, few have focused on the actual role and impact of architecture parameters within DARTS. This gap in understanding gives rise to critical ambiguities in the architecture search process: Are architecture parameters truly necessary for architecture se-

lection in DARTS frameworks? What is their actual role and impact? How can we better utilize these parameters to develop more effective differentiable NAS methodologies?

To address this gap, we empirically demonstrate that architecture parameters are indispensable for architecture selection in the DARTS framework. Through rigorous theoretical analysis, we reveal that the change rate of architecture parameters reflects the sensitivity of the supernet's validation loss in architecture space, influencing the performance of the derived architecture by shaping the dynamics of supernet training. These insights help resolve critical ambiguities surrounding the actual role and influence of architecture parameters in the DARTS framework in existing DARTS-related research. Building on this foundation, we introduce the concept of the 'Stable Equilibrium State', which offers essential insights into the validation loss trajectory across architecture spaces and elucidates the stability of the supernet's bi-level optimization process. We further investigate the supernet training dynamics to elucidate the influence of operations on the Stable Equilibrium State, subsequently leading to the proposal of a novel metric for evaluating operation importance, termed Equilibrium Influential ($E_\mathcal{I}$). Through theoretical validation, we demonstrate that $E_\mathcal{I}$ reliably reflects the true significance of operations within the architecture. Integrating these elements, we introduce BOSE-NAS, a differentiable NAS method that utilizes the Stable Equilibrium State to identify the optimal state during the search process, subsequently deriving the final architecture based on the $E_\mathcal{I}$ metric. Extensive experiments conducted on different datasets across various search spaces demonstrate its effectiveness and efficiency. In the DARTS search space, BOSE-NAS achieves an impressive average test error of $2.49\%$ and a best test error of $2.37\%$ on the CIFAR-10 dataset. When transferred to CIFAR-100 and ImageNet, BOSE-NAS attains an average test error of $16.23\%$ and a best test error of $16.08\%$ on CIFAR-100, and a best test error of $24.1\%$ on ImageNet. Remarkably, our method accomplishes this with a mere 0.13 GPU-days of computational cost (equivalent to just 3 hours of search time on a single V100 GPU) for architecture search on CIFAR-10. This level of efficiency outperforms DARTS by more than 3 times and surpasses DARTS-PT by nearly 6 times.

In summary, our contributions are as follows:

- We provide comprehensive empirical and theoretical analyses to elucidate the actual role and impact of architecture parameters $\alpha$ within the DARTS framework, addressing a critical gap in the existing literature.
- We introduce the concept of the Stable Equilibrium State, which offers essential insights into the stability of the supernet's bi-level optimization process. Additionally, we propose Equilibrium Influential ($E_\mathcal{I}$), a novel and robust metric for evaluating the importance of operations.
- We present an innovative and effective differentiable NAS method, termed BOSE-NAS, and demonstrate its superior performance and search efficiency through extensive experimentation across a variety of datasets and search spaces.

## 2 RELATED WORKS

NAS-RL (Zoph & Le, 2016) and MetaQNN (Baker et al., 2022) are pioneering methods in the field of neural architecture search (NAS). These studies employed reinforcement learning (RL) methods to design neural architectures that achieved state-of-the-art classification accuracy on image classification tasks, thereby demonstrating the feasibility of automated neural architecture design. Following this, AmoebaNet (Real et al., 2019) further validated the concept by employing an evolutionary algorithm to achieve similar results. However, these methods required significant computational resources, often consuming hundreds of GPU days or more.

To mitigate the issue of high computational costs and expedite the neural architecture search process, (Liu et al., 2018b) proposed the Differentiable Architecture Search (DARTS). DARTS is a widely adopted one-shot method that facilitates the efficient exploration of architectures through gradient descent by employing continuous relaxation. It transforms discrete architecture selection into continuous parameters $\alpha$, optimizes architecture parameters via gradient descent, and subsequently constructs the final architecture by selecting the operations associated with the highest parameter values.

Despite its efficiency, DARTS has encountered significant challenges related to performance degradation (Liang et al., 2019; Wang et al., 2021; Zela et al., 2019). Numerous studies have shown that

architecture parameters are negatively impacted during supernet training, leading to various proposals aimed at controlling or adjusting these parameters to more accurately reflect the true strengths of operations. For instance, RobustDARTS (Zela et al., 2019) demonstrates that low curvature (eigenvalues of the Hessian matrix of the validation loss) does not cause significant performance drops and proposes an early stopping criterion by monitoring these eigenvalues. SDARTS (Chen & Hsieh, 2020) addresses inaccuracies in gradient computation of architecture parameters, which creates a significant optimization gap and introduces a method to amend architecture gradients to reduce this gap. Additionally, Yang et al. (Huang et al., 2020) proposed EnTranNAS, a heuristic method that assesses validation loss in sub-networks by iteratively evaluating Engine-Cells and Transit-Cells, where Engine-Cells are differentiable for architecture search and Transit-Cells facilitate the subgraph transition. SGAS (Li et al., 2020) introduced a sequential greedy architecture search method incorporating heuristic criteria such as edge importance, selection certainty, and stability to mitigate the search-evaluation correlation issue. Finally, Huang et al. (Huang et al., 2020) suggest reducing the evaluation gap by introducing a collection of topological variables with a combinatorial probabilistic distribution to explicitly model the desired topology.

Conversely, a substantial body of recent literature indicates that the limitations of DARTS primarily stem from the architecture parameters' inability to accurately reflect the true strength of operations, prompting the development of new metrics for evaluating operation significance. DARTS-PT (Wang et al., 2021) mathematically demonstrates the intrinsic phenomenon of skip connection dominance, leading to performance collapse, and introduces a perturbation-based architecture selection method where operation strength is gauged by its impact on supernet accuracy. EoiNAS (Zhou et al., 2021) utilizes the ratio of training iterations to validation accuracy for selecting the final operations. Shapley-NAS (Xiao et al., 2022) quantifies the marginal contribution of operations on accuracy using Shapley values, approximated through Monte Carlo sampling. DARTS-IM (Zhang et al., 2022) reveals that operation strength depends on both magnitude and second-order information, and introduces Influential Magnitude, a new criterion that incorporates this information for operation selection.

Unlike previous research, this paper focuses on investigating the actual role and impact of architecture parameters within DARTS. By filling this significant gap, we aim to propose a more effective differentiable NAS method.

## 3 APPROACH

In this section, we conduct comprehensive empirical and theoretical analyses of the role and impact of architecture parameters $\alpha$ within the DARTS framework. Building on this foundation, we propose an innovative and effective differentiable NAS method.

### 3.1 PRELIMINARIES: DARTS AND THE BI-LEVEL OPTIMIZATION

DARTS is one of the most popular solutions to identify effective architectures, as it largely reduces the search cost by relaxing the architecture search to continuous mixture weights learning. Following prior works (Liu et al., 2018a; Real et al., 2019; Zoph et al., 2018), DARTS searches for the best cell structure and constructs the supernet by repetitions of normal and reduction cells. Each cell is represented as a directed acyclic graph (DAG) comprising $N$ nodes, where each node represents a latent feature. Each edge $(i, j)$ includes multiple candidate operations. DARTS applies continuous relaxation to integrate the results of candidate operations, whose strength is measured by architecture parameters denoted as $\alpha$.

$$\beta_k^{(i,j)} = \frac{\exp\left(\alpha_k^{(i,j)}\right)}{\sum_{k'=1}^{|\mathcal{O}|} \exp\left(\alpha_{k'}^{(i,j)}\right)} \tag{1}$$

where $O$ is the set of all candidate operations, $\beta$ is the softmax-activated set of architecture parameters $\alpha$. DARTS utilizes a bi-level optimization framework to iteratively optimize the architecture parameters $\alpha$ and model weights $\omega$:

$$\min_{\alpha} \mathcal{L}_{\text{valid}}\left(\alpha, \omega^*(\alpha)\right) \tag{2}$$

$$s.t. \omega^*(\alpha) = \arg\min_{\omega} \mathcal{L}_{\text{train}}\left(\alpha, \omega\right) \tag{3}$$

where $\mathcal{L}_{\text{train}}$ and $\mathcal{L}_{\text{valid}}$ are the validation and training loss, respectively. The goal for architecture search is to find $\alpha^*$ to minimize the validation loss $\mathcal{L}_{\text{valid}}$ and $\omega^*$ is obtained by minimizing the training loss $\mathcal{L}_{\text{train}}$. Among them, by setting the evaluation point $\omega^{'} = \omega - \xi\nabla_\omega\mathcal{L}_{\text{train}}(\alpha,\omega)$, the total derivative of $\mathcal{L}_{\text{valid}}$ w.r.t. $\alpha$ evaluated on $(\alpha, \omega^*(\alpha))$ would be:

$$\frac{d\mathcal{L}_{\text{valid}}}{d\alpha}(\alpha) = \nabla_\alpha\mathcal{L}_{\text{valid}}\left(\alpha,\omega^{'}\right) - \xi\nabla_{\omega^{'}}\mathcal{L}_{\text{valid}}\left(\alpha,\omega^{'}\right)\nabla^2_{\alpha,\omega}\mathcal{L}_{\text{train}}(\alpha,\omega) \tag{4}$$

Utilizing the finite difference approximation around $\omega^{\pm} = \omega \pm \epsilon\nabla_{\omega^{'}}\mathcal{L}_{\text{valid}}\left(\alpha,\omega^{'}\right)$ for small $\epsilon = 0.01/\left\|\nabla_\omega\mathcal{L}_{\text{valid}}\left(\alpha,\omega^{'}\right)\right\|_2$ to reduce the complexity, Equation 4 can be rewritten as:

$$\frac{d\mathcal{L}_{\text{valid}}}{d\alpha}(\alpha) = \nabla_\alpha\mathcal{L}_{\text{valid}}\left(\alpha,\omega^{'}\right) - \frac{\xi}{2\epsilon}\left(\nabla_\alpha\mathcal{L}_{\text{train}}\left(\alpha,\omega^{+}\right) - \nabla_\alpha\mathcal{L}_{\text{train}}\left(\alpha,\omega^{-}\right)\right) \tag{5}$$

To further accelerate the optimization process, DARTS employs a first-order approximation by setting $\xi = 0$. This simplification effectively eliminates the second-order derivative in Equation 4 and its corresponding approximation in Equation 5.

At the end of the training phase, operations associated with the largest architecture parameter on each edge will be selected from the supernet to construct the final architecture.

## 3.2 THE ROLE AND IMPACT OF ARCHITECTURE PARAMETER

Despite its efficiency, DARTS has faced several challenges related to performance degradation issues (Liang et al., 2019; Wang et al., 2021; Zela et al., 2019; Zhang et al., 2022; Xue et al., 2022). To address these challenges, existing literature has predominantly focused on either adjusting architecture parameters to more accurately reflect the true strengths of operations (Ye et al., 2022; Chu et al., 2020b;a) or developing alternative metrics for evaluating operation significance (Wang et al., 2021; Xiao et al., 2022; He et al., 2024). However, there has been limited investigation into the actual role and impact of architecture parameters within the DARTS framework. This gap in understanding introduces critical ambiguities in the architecture search process.

To address this significant gap, we undertake comprehensive empirical and theoretical analyses. We contend that although architecture parameters may not directly represent operation significance, they significantly influence the architecture search process and thus affect the performance of the derived architecture. To substantiate this claim, we conduct empirical analyses in the NAS-Bench-201 search space (Dong & Yang, 2020). We train two sets of supernets, with one set having architecture parameter $\alpha$ fixed. Each set of supernets is trained on CIFAR-10 and CIFAR-100 using three different seeds. We apply two well-known architecture selection methods, DARTS-PT and RMI-NAS, to derive an architecture from the supernets every 10 epochs and record their stand-alone model accuracy. The results, as illustrated in Figure 1, reveal a notable discrepancy in the test accuracies of architectures derived from supernets with fixed versus unfixed $\alpha$, indicating that $\alpha$ indeed impacts performance, likely mediated through supernet training.

To understand the actual role and impact of architecture parameters, we theoretically analyze the bi-level optimization process during architecture search. Based on Equation 5, we perform a Taylor expansion on $\alpha$ for validation loss:

$$\frac{d\mathcal{L}_{\text{valid}}}{d\alpha}(\alpha) = \frac{\Delta\mathcal{L}_{\text{valid}}\left(\alpha,\omega^{'}\right)}{\Delta\alpha_\varepsilon} - \frac{\xi}{2\epsilon}\left(\frac{\Delta\mathcal{L}_{\text{train}}\left(\alpha,\omega^{+}\right)}{\Delta\alpha_\varepsilon} - \frac{\Delta\mathcal{L}_{\text{train}}\left(\alpha,\omega^{-}\right)}{\Delta\alpha_\varepsilon}\right) + o\left(\Delta\alpha_\varepsilon\right) \tag{6}$$

Here, $\xi$ represents the learning rate of the weight parameters $\omega$, $\epsilon$ is a small scalar dependent on $\omega$, and $\Delta\alpha_\varepsilon$ denotes the change in the architecture parameter $\alpha$ between time steps $t$ and $t + \varepsilon$, where $\varepsilon$ is an infinitesimal scalar related on $t$. In this paper, we adopt the first-order architecture gradient approximation as proposed in the original DARTS Liu et al. (2018b). Consequently, $\mathcal{L}_{\text{train}}\left(\alpha,\omega^{+}\right)$ and $\mathcal{L}_{\text{train}}\left(\alpha,\omega^{-}\right)$ have little effect on $\alpha$. Let $\alpha_t$ be the architecture parameter with $\Delta_t$ epoch updates from the initial $\alpha_0$:

$$\alpha_t = \alpha_0 - \eta\Delta t\frac{d\mathcal{L}_{\text{valid}}}{d\alpha}(\alpha) \tag{7}$$

where $\eta$ is the learning rate of $\alpha$. We now have:

$$\frac{\Delta\mathcal{L}_{valid}\left(\alpha,\omega^{'}\right)}{\Delta\alpha_\varepsilon} \approx \frac{1}{\eta}\frac{\Delta\alpha_t}{\Delta t} \tag{8}$$

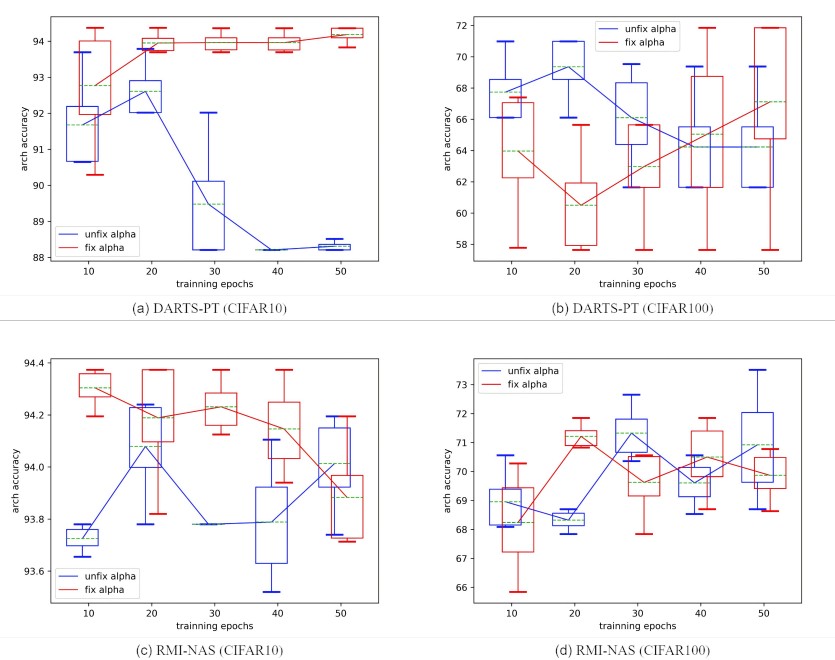

Figure 1: Analysis of the impact of architecture parameters on the accuracy of architectures discovered by DARTS-PT and RMI-NAS on CIFAR-10 and CIFAR-100 datasets. Blue bars represent training the supernet using the unfixed $\alpha$ method, while red bars indicate training with the fixed $\alpha$ method. Panels (a) and (b) display the results for DARTS-PT on CIFAR-10 and CIFAR-100, respectively, whereas panels (c) and (d) present the results for RMI-NAS on CIFAR-10 and CIFAR-100.

where $\Delta\alpha_t = \alpha_0 - \alpha_t$. The detailed theoretical proof process is provided in Appendix A.1

Based on the aforementioned analyses, we conclude the precise role and impact of architecture parameters in DARTS: the change rate of architecture parameters actually reflects the sensitivity of the supernet's validation loss in architecture space, influencing the performance of the derived architecture by shaping the dynamics of supernet training.

### 3.3 A DIFFERENTIABLE NEURAL ARCHITECTURE SEARCH WITH BI-LEVEL OPTIMIZATION STABLE EQUILIBRIUM

Drawing on insights from Section 3.2, we observe that during the supernet's bi-level optimization process, if the sensitivity of the validation loss to changes in $\alpha$ fluctuates greatly over time, it suggests that the supernet optimization is highly sensitive to small perturbations in architecture parameters. Conversely, if this sensitivity remains low, it indicates that the training process is approaching a relatively stable state. To formalize this, we introduce the concept of the "Stable Equilibrium State," defined in Equation 9 and visually demonstrated in Figure 2. The Stable Equilibrium State provides essential insights into the validation loss trajectory across architecture spaces.

$$\left|\frac{\Delta s}{\Delta t}\right| \approx \left|\frac{1}{\eta_2}\frac{\Delta\rho}{\Delta t}\right| \tag{9}$$

where $\rho = \left|\frac{\Delta\alpha}{\Delta t}\right|$, $s = \left|\frac{\Delta\mathcal{L}_{\text{valid}}\left(\alpha,\omega'\right)}{\Delta\alpha_\varepsilon}\right|$.

Moreover, we further investigate the dynamics of the supernet training process to elucidate the influence of operations on the Stable Equilibrium State. Consequently, we introduce Equilibrium Influential ($E_{\mathcal{I}}$), a novel metric designed to assess the significance of operations. The metric is

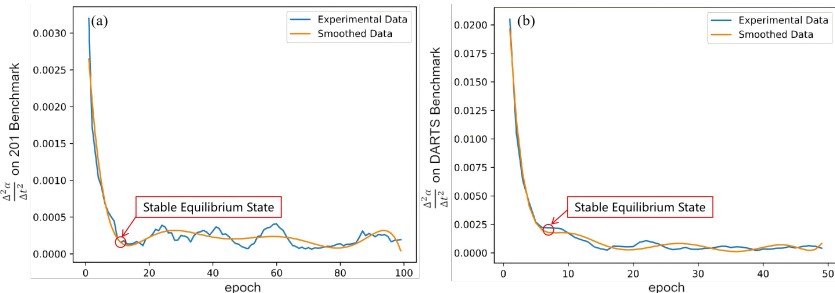

Figure 2: The second-order differential of the architecture parameters over time on the CIFAR-10 dataset across (a) 201 benchmark and (b) DARTS benchmark. The blue line represents the experimental data, and the orange line represents the smoothed data. The red circle represents the first Stable Equilibrium State.

formally defined as:

$$E_{\mathcal{I}} = \text{Sigmoid} \left( \frac{\Delta\alpha_{t+\Delta t} - \Delta\alpha_t}{\Delta t^2} + \frac{\Delta\alpha_t - \Delta\alpha_{t-\Delta t}}{\Delta t^2} + \frac{\Delta\alpha_{t-\Delta t} - \Delta\alpha_{t-2\Delta t}}{\Delta t^2} \right) \tag{10}$$
$$+ \frac{\alpha_{t+\Delta t} - \alpha_t}{\Delta t} + \frac{\alpha_t - \alpha_{t-\Delta t}}{\Delta t} + \frac{\alpha_{t-\Delta t} - \alpha_{t-2\Delta t}}{\Delta t}$$

The proposed $E_{\mathcal{I}}$ metric encapsulates both the first and second differential of the architecture parameters across three distinct time points, thus providing a comprehensive measure of their dynamic impact on sensitivity and the Stable Equilibrium State of the validation loss. We employ a sigmoid function to mitigate the sensitivity of the second derivatives of the architecture parameters to extreme variations, reducing the influence of outliers and noise in the data, and thereby enhancing the metric's robustness.

We leverage the Influence Function (Meng et al., 2020) to theoretically validate the reliability of the $E_{\mathcal{I}}$ metric as a measure of operation importance. By conducting Taylor expansion on $\frac{\Delta\bar{\alpha}}{\Delta\bar{t}}$ for Sigmoid $\left( \frac{\Delta\alpha_{t+\Delta t} - \Delta\alpha_{t-2\Delta t}}{\Delta t^2} \right)$, we can approximate Equation 10 as:

$$E_{\mathcal{I}} \approx \frac{1}{1 + e^{-\frac{\Delta\bar{\alpha}}{\Delta\bar{t}}}} + \frac{e^{-\frac{\Delta\bar{\alpha}}{\Delta\bar{t}}}}{\left(1 + e^{-\frac{\Delta\bar{\alpha}}{\Delta\bar{t}}}\right)^2} \left( -\frac{\Delta\bar{\alpha}}{\Delta\bar{t}} \right) + \left( 3\frac{\Delta\bar{\alpha}}{\Delta\bar{t}} \right) \tag{11}$$

Among them, $\frac{\Delta\bar{\alpha}}{\Delta\bar{t}} = \frac{\alpha_{t+\Delta t} - \alpha_{t-2\Delta t}}{3\Delta t}$, function $f(x) = \frac{1}{1+e^{-x}} - \frac{e^{-x}}{(1+e^{-x})^2}x + 3x$ is monotonically increasing when $x$ is between $-1$ and $1$. Hence, we can obtain that $E_{\mathcal{I}}$ positively correlates with $\frac{\Delta\bar{\alpha}}{\Delta\bar{t}}$. As (Meng et al., 2020; Koh & Liang, 2017) state that the influence on validation loss (denoted as $I(\theta, L)$) is positively correlated with the derivatives of validation loss w.r.t $\alpha$:

$$I(\theta, \mathcal{L}) \propto \nabla_{\alpha}\mathcal{L}(\theta, \alpha) \tag{12}$$

Combining the findings above, we derive:

$$E_{\mathcal{I}} \propto I(\theta, \mathcal{L}(\theta, \alpha)) \tag{13}$$

The theoretical proof for the above derivation is provided in Appendix A.2.

Integrating our findings, we introduce BOSE-NAS, a differentiable neural architecture search method based on the Stable Equilibrium State of the bi-level optimization. BOSE-NAS utilizes the Stable Equilibrium State to identify the optimal state of the search process, subsequently deriving the final architecture based on the $E_{\mathcal{I}}$ metric. Our methodology is summarized in Algorithm 1.

Notably, in our approach, we track the Stable Equilibrium State from the beginning of supernet training and stop training upon encountering the first minima, which implies that the supernet has reached relative stability. While multiple local minima may exist with extended training as shown in Figure 2, we prioritize the supernet at the first one to derive the final architecture. The rationale behind this approach is that ignoring the first local minimum in favor of subsequent ones or

pursuing a global minimum could risk overfitting, ultimately impairing performance, as evidenced by references (Liang et al., 2019; Zela et al., 2019; Chen et al., 2021b). To support our strategy of designating the first Stable Equilibrium State for architecture derivation, we conducted an empirical analysis in the NAS-Bench-201 and DARTS search spaces. As part of this analysis, we identified architectures corresponding to each local minimum encountered during the training phase and evaluated their performance. The results, detailed in the Ablation Study section, confirm that the architecture derived from the first local minimum outperforms the others, consistent with our previous observations.

## 4 EXPERIMENTS

In this section, we evaluate the effectiveness of BOSE-NAS across various search spaces and widely used image classification benchmark datasets. The results highlight the strong competitiveness of BOSE-NAS when compared to other state-of-the-art search methodologies.

### 4.1 SEARCH SPACE AND DATASET

To evaluate the efficacy of our approach, we conducted comprehensive experiments across several prevalent search spaces and datasets commonly used in differentiable architecture searches. The explored search spaces include the DARTS search space (Liu et al., 2018b), NAS-Bench-201 search space (Dong & Yang, 2020), as well as the S1-S4 search spaces (Zela et al., 2019). The datasets include CIFAR-10, CIFAR-100, and ImageNet, ensuring a thorough evaluation of our method's performance across diverse domains.

### 4.2 IMPLEMENTATION DETAILS

Our experimental setup for search and evaluation follows standard research practices. In the DARTS and S1-S4 search spaces, the supernet comprises 6 normal cells and 2 reduction cells, with each cell containing 6 nodes. In the NAS-Bench-201 search space, the supernet consists of 15 normal cells, with each cell containing 4 nodes. During super-net training, we set the number of epochs to 50 for the DARTS search space, 100 for the NAS-Bench-201 search space, and 20 for the S1-S4 search space, Use SGD with an initial learning rate of 0.025, momentum of 0.9, batch size of 64, and weight decay of $3 \times 10^{-4}$ to optimize the supernet weights. During the architecture retraining phase, an architecture with 18 normal cells and 2 reduction cells is retrained from scratch on CIFAR-10/100. The architecture is optimized by the SGD optimizer with an initial learning rate of 0.025, momentum of 0.9, drop path rate of 0.2, weight decay of $3 \times 10^{-4}$, and gradient clipping at 5 for 600 epochs. An architecture with 12 normal cells and 2 reduction cells is retrained from scratch on ImageNet. The architecture is optimized by the SGD optimizer with an initial learning rate of 0.4, momentum of 0.9, drop path rate of 0.2, weight decay of $3 \times 10^{-5}$, and gradient clipping at 5 for 250 epochs. The hyperparameter $\Delta t$ determines the smoothness of the second-order differential of the architecture parameters and the absolute value of the proposed operation importance metric in our method. However, it does not change the trajectory or the relative importance of the operations.

### 4.3 ARCHITECTURE SEARCH AND EVALUATION ON CIFAR-10

In the CIFAR-10 dataset, we extensively assessed the BOSE-NAS method across multiple search spaces. In the DARTS search space, supernet training was set for 50 iterations, with $\Delta t$ set to 5. Similarly, we determined the final network architecture using the BOSE-NAS method. We conducted three searches using different random seeds. Following the procedure described in Section 4.2, the architectures were retrained, with the results presented in Table 1. The architectures we obtained had an average error rate of 2.49% and a best error rate of 2.37%. While the average accuracy was slightly lower than that of OLES, it surpassed all other methods. The best accuracy also exceeded that of all the methods compared. Regarding the S1-S4 search spaces, we set the number of iterations for supernet training to 20, with the parameter $\Delta t$ set to 2. Three network architectures were searched using different random seeds. These architectures were then retrained following the methodology outlined in Section 4.2, and the results are shown in Table 3. Our method achieved an average test accuracy of 97.27% on the S1 search space, 97.45% on the S2 search space, and 97.47% on the S4 search space, outperforming all state-of-the-art (SOTA) methods in comparison.

Table 1: Comparison with state-of-the-art methods on CIFAR-10 and CIFAR-100.

| Architectures | Test Error (%) | | Params (M) | Search Cost (GPU-days) | Type |
|---|---|---|---|---|---|
| | CIFAR-10 | CIFAR-100 | | | |
| DenseNet-BC (Huang et al., 2017) | 3.46 | 17.18 | 25.6 | - | manual |
| AmoebaNet-B (Real et al., 2019) | 2.55±0.05 | - | 2.8 | 3150 | evolution |
| ENAS (Pham et al., 2018) | 2.89 | - | 4.6 | 0.5 | RL |
| NASNet-A (Zoph et al., 2018) | 2.65 | - | 3.3 | 1800 | RL |
| PNAS (Liu et al., 2018a) | 3.41±0.09 | - | 3.2 | 225 | SMBO |
| DARTS (1st)(2018) (Liu et al., 2018b) | 3.00±0.14 | - | 3.3 | 0.4 | gradient |
| DARTS (2nd) (Liu et al., 2018b) | 2.76±0.09 | - | 3.3 | 1 | gradient |
| SNAS(2018) (Xie et al., 2019) | 2.85±0.02 | - | 2.8 | 1.5 | gradient |
| P-DARTS(2019) (Chen et al., 2019b) | 2.50 | 16.55 | 3.4 | 0.3 | gradient |
| PC-DARTS(2019) (Xu et al., 2019) | 2.57±0.07 | 15.92 | 3.6 | 0.1 | gradient |
| GDAS(2019) (Dong & Yang, 2019) | 2.82 | 18.13 | 2.5 | 0.17 | gradient |
| DropNAS(2020) (Hong et al., 2020) | 2.58±0.14 | 16.95±0.41 | 4.1 | 0.6 | gradient |
| FairDARTS(2020) (Chu et al., 2020b) | 2.54 | - | 2.8 | 0.4 | gradient |
| DARTS-PT(2021) (Wang et al., 2021) | 2.61±0.08 | - | 3.0 | 0.8 | gradient |
| EoiNAS(2021) (Zhou et al., 2021) | 2.50 | 17.3 | 3.4 | 0.6 | gradient |
| $\beta$-DARTS(2022) (Ye et al., 2022) | 2.51±0.08 | 16.52±0.03 | 3.8 | 0.4 | gradient |
| Zero-Cost-PT(2023) (Xiang et al., 2023) | 2.62 | - | 4.6 | 0.17 | gradient |
| OLES(2023) (Jiang et al., 2023) | 2.41±0.11 | 17.30 | 3.4 | 0.4 | gradient |
| IS-DARTS(2024) (He et al., 2024) | 2.56±0.04 | - | 4.25 | 0.42 | gradient |
| BOSE-NAS (Avg) | 2.49±0.11 | 16.23±0.11 | 4.24 | 0.13 | gradient |
| **BOSE-NAS (Best)** | 2.37 | 16.08 | 4.24 | 0.13 | gradient |

Table 2: Comparison with state-of-the-art method on ImageNet.

| Architecture | Test Error (%) | Params (M) |
|---|---|---|
| Inception-v1 (Szegedy et al., 2015) | 30.1 | 6.6 |
| MobileNet (Howard et al., 2017) | 29.4 | 4.2 |
| NASNet-A (Zoph et al., 2018) | 26.0 | 5.3 |
| AmoebaNet-C (Real et al., 2019) | 24.3 | 6.4 |
| PNAS (Liu et al., 2018a) | 25.8 | 5.1 |
| DARTS (2nd)(2018) (Liu et al., 2018b) | 26.7 | 4.7 |
| SNAS (mild)(2018) (Xie et al., 2019) | 27.3 | 4.3 |
| GDAS(2019) (Dong & Yang, 2019) | 26.0 | 5.3 |
| P-DARTS(2019) (Chen et al., 2019b) | 24.4 | 4.9 |
| PC-DARTS(2019) (Xu et al., 2019) | 25.1 | 5.3 |
| SGAS(Cri 1. best)(2019) (Li et al., 2020) | 24.2 | 5.3 |
| DrNAS(2020) (Chen et al., 2021a) | 24.2 | 5.2 |
| DARTS-PT(2021) (Wang et al., 2021) | 25.5 | 4.7 |
| EoiNAS(2021) (Zhou et al., 2021) | 25.6 | 5.0 |
| $\beta$-DARTS(2022) (Ye et al., 2022) | 23.9 | 5.5 |
| Zero-Cost-PT(2023) (Xiang et al., 2023) | 24.4 | 6.3 |
| OLES(2023) (Jiang et al., 2023) | 24.7 | 4.7 |
| IS-DARTS(2024) (He et al., 2024) | 24.1 | 6.4 |
| **BOSE-NAS (Best)** | 24.1 | 5.9 |

However, in the S3 search space, our method achieved an average accuracy of $97.47\%$, which is slightly lower than that of PC-DARTS, DARTS-PT, and Shapley-NAS. These results indicate that BOSE-NAS can explore competitive network architectures in the S1-S4 search spaces. In addition, in the NAS-Bench-201 search space, we set the number of iterations for supernet training to $100$, with the parameter $\Delta t$ set to $10$. Repeated experiments with different random seeds, as shown in Table 4, confirmed that our method effectively explores optimal network architectures on CIFAR-10.

Combining all experimental results, we concluded that the BOSE-NAS method effectively identifies superior network architectures across various search spaces, demonstrating significant competitiveness compared to architectures derived by other methods. This confirms that our approach is feasible and effective. Notably, our method can identify the optimal state at the early stage of supernet training for architecture derivation, thereby avoiding a converged but overfitted supernet with deteriorated performance, as empirically demonstrated in (Zela et al., 2019; Liang et al., 2019). Moreover, unlike some previous methods (Chen et al., 2019a; Hong et al., 2020; Wang et al., 2021; Li et al., 2019) that involve high computational complexity for assessing operation strength, the proposed $E_{\mathcal{I}}$ metric operates with lower overhead while maintaining reliable performance. Therefore, with the early identification of a stable supernet and an efficient and reliable operation evaluation, our approach offers significant improvements in test accuracy and substantial reductions in search cost. In the DARTS search space, our approach on CIFAR-10 requires only $0.13$ GPU-days for the search. This efficiency surpasses DARTS by over threefold and outperforms DARTS-PT by nearly sixfold.

## 4.4 ARCHITECTURE TRANSFERABILITY EVALUATION

To evaluate the generalization capability of our proposed method, we transferred architectures discovered in the DARTS space on the CIFAR-10 dataset to the CIFAR-100 and ImageNet datasets. For CIFAR-100, we adopted the same retraining model architecture for CIFAR-10, comprising $18$ normal cells and $2$ reduction cells. This architecture was trained for $600$ epochs with an initial learn-

Table 3: Comparison with state-of-the-art method on S1, S2, S3, and S4 search space.

| Architectures | CIFAR-10 | | | |
|---|---|---|---|---|
| | S1 | S2 | S3 | S4 |
| DARTS(2018) (Liu et al., 2018b) | 3.84 | 4.85 | 3.34 | 7.2 |
| PC-DARTS(2019) (Xu et al., 2019) | 3.11 | 3.02 | 2.51 | 3.02 |
| R-DARTS(2019) (Zela et al., 2019) | 3.11 | 3.48 | 2.93 | 3.58 |
| DARTS-(2020) (Chu et al., 2020a) | 2.76±0.07 | 2.79±0.04 | 2.65±0.04 | 2.91±0.04 |
| SDARTS(2020) (Chen & Hsieh, 2020) | 2.78 | 2.75 | 2.53 | 2.93 |
| DARTS-PT(2021) (Wang et al., 2021) | 3.5 | 2.79 | 2.49 | 2.64 |
| Shapley-NAS(2022) (Xiao et al., 2022) | 2.82 | 2.55 | 2.42 | 2.63 |
| BOSE-NAS (Avg) | 2.73±0.11 | 2.55±0.07 | 2.53±0.02 | 2.53±0.11 |
| **BOSE-NAS (Best)** | 2.62 | 2.45 | 2.52 | 2.49 |

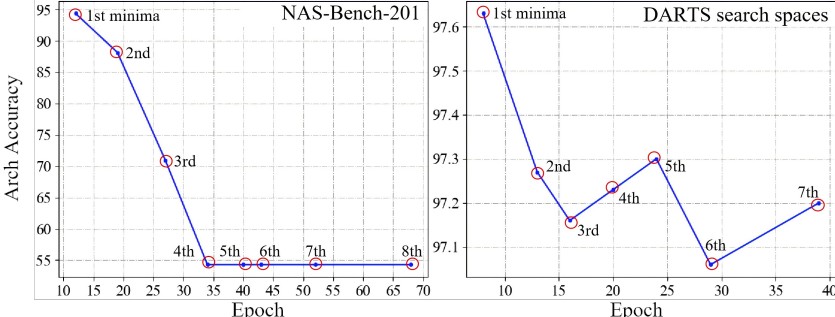

Figure 3: The performance of architectures derived at different minima in NAS-Bench-201(left) and DARTS search space(right).

ing rate of 0.025. As shown in Table 1, experimental validation yielded an average error rate of 16.23% and a best error rate of 16.08% on CIFAR-100. Although the accuracy was slightly lower than that of PC-DARTS, it surpassed all other SOTA methods, showcasing significant competitiveness compared to prior methods. On the ImageNet dataset, we constructed a new model architecture consisting of 14 normal cells and 2 reduction cells. The training was conducted for 250 epochs using four V100 GPUs with an initial learning rate of 0.4. As shown in Table 2, the results indicated the best error rate of 24.1% on ImageNet, an accuracy slightly lower than that of $\beta$-DARTS but surpassing all other SOTA methods in comparison. Comparative analysis with other methods confirmed the efficacy of our architecture search approach. In summary, we successfully transferred architectures discovered on CIFAR-10 to CIFAR-100 and ImageNet, affirming their excellent generalization capability. Our approach exhibits robust competitiveness when compared to alternative methods.

## 4.5 ABLATION STUDY

To validate our strategy of designating the first Stable Equilibrium State for architecture derivation, we conducted ablation studies to evaluate the performance of architectures derived from the supernet at various minima on the CIFAR-10 dataset in the NAS-Bench-201 and DARTS search space. The result is shown in Figure 3. In NAS-Bench-201, the local minima were observed sequentially at epochs 12, 18, 27, 34, 40, 43, 52, and 68 during the training process. Notably, the architecture derived at the first minima achieved an optimal accuracy of 94.37%. In contrast, the accuracies at subsequent minima were significantly lower. Similarly, in the DARTS search spaces, the architecture derived from the first minima outperformed those derived later. Therefore, we consider it reasonable to designate the first Stable Equilibrium State for deriving the final architecture.

## 5 LIMITATION

Our operation importance metric evaluates the relative significance of operations by independently assessing their influence on supernet stability. However, it does not account for the intricate dependencies between operations, which is a limitation of our current approach. Additionally, since the metric is based on the Stable Equilibrium State identified in our method, it may not be directly applicable to other DARTS methodologies.

Table 4: Comparison with state-of-the-art method on NAS-Bench-201. The results in parentheses represent the upper bound.

| Method | CIFAR-10 | | CIFAR-100 | | ImageNet-16-120 | |
|---|---|---|---|---|---|---|
| | validation | test | validation | test | validation | test |
| ResNet (He et al., 2016) | 90.83 | 93.97 | 70.42 | 70.86 | 44.53 | 43.63 |
| Random | 90.93±0.36 | 93.70±0.36 | 70.60±1.37 | 70.65±1.38 | 42.92±2.00 | 90.93±2.15 |
| ENAS (Pham et al., 2018) | 39.77±0.00 | 54.30±0.00 | 10.23±0.12 | 10.62±0.27 | 16.43±0.00 | 16.32±0.00 |
| DARTS(2018) (Liu et al., 2018b) | 39.77±0.00 | 54.30±0.00 | 15.03±0.00 | 15.61±0.00 | 16.43±0.00 | 16.32±0.00 |
| SNAS(2018) (Xie et al., 2019) | 90.10±1.04 | 92.77±0.83 | 69.69±2.39 | 69.34±1.98 | 42.84±1.79 | 43.16±2.64 |
| GDAS(2019) (Dong & Yang, 2019) | 90.01±0.46 | 93.23±0.23 | 24.05±8.12 | 24.20±8.08 | 40.66±0.00 | 41.02±0.00 |
| PC-DARTS(2019) (Xu et al., 2019) | 89.96±0.15 | 93.41±0.30 | 67.12±0.39 | 67.48±0.89 | 40.83±0.08 | 41.31±0.22 |
| DrNAS(2020) (Chen et al., 2021a) | 91.55±0.00 | 94.36±0.00 | 73.49±0.00 | 73.51±0.00 | 46.37±0.00 | 46.34±0.00 |
| IDARTS(2021) (Zhang et al., 2021b) | 89.96±0.60 | 93.58±0.32 | 70.57±0.24 | 70.83±0.48 | 40.38±0.59 | 40.89±0.68 |
| DARTS-PT(2021) (Wang et al., 2021) | - | 88.11 | - | - | - | - |
| DARTS-PT(fix alpha) (Wang et al., 2021) | - | 93.80 | - | - | - | - |
| DARTS-IM(2022) (Zhang et al., 2022) | - | 93.61±0.23 | - | 71.31±0.40 | - | 44.98±0.36 |
| $\beta$-DARTS(2022) (Ye et al., 2022) | 91.55±0.00 | 94.36±0.00 | 73.49±0.00 | 73.51±0.00 | 46.37±0.00 | 46.34±0.00 |
| OLES(2023) (Jiang et al., 2023) | 90.88±0.10 | 93.70±0.15 | 70.56±0.28 | 70.40±0.22 | 44.17±0.49 | 43.97±0.38 |
| IS-DARTS(2024) (He et al., 2024) | 91.55±0.00 | 94.36±0.00 | 73.49±0.00 | 73.51±0.00 | 46.37±0.00 | 46.34±0.00 |
| BOSE-NAS (Avg) | 91.42±0.13 | 94.21±0.22 | 72.78±0.91 | 72.72±0.51 | 45.52±0.10 | 46.27±0.08 |
| **BOSE-NAS (Best)** | 91.50 | 94.37 | 73.31 | 73.09 | 45.58 | 46.63 |

## 6 CONCLUSION

This paper addresses a critical gap in existing DARTS-related research by investigating the actual role and impact of architecture parameters in the DARTS and proposing a more effective differentiable NAS method. We empirically demonstrate that architecture parameters are indispensable for architecture selection in the DARTS framework. Through rigorous theoretical analysis, we uncover their true significance, resolving longstanding ambiguities in the interpretation of architecture parameters in prior research. Building on these insights, we introduce the concept of the 'Stable Equilibrium State', which provides crucial insights into the validation loss trajectory across architecture spaces. Further exploration of supernet training dynamics reveals the influence of operations on the Stable Equilibrium State during training, leading us to propose a novel metric, the Equilibrium Influential ($E_{\mathcal{I}}$) metric, to quantify the significance of operations. By integrating these elements, we introduce BOSE-NAS, a novel differentiable NAS method that utilizes the Stable Equilibrium State to identify the optimal state of the search process and subsequently derives the final architecture based on the $E_{\mathcal{I}}$ metric. The effectiveness of BOSE-NAS is demonstrated through significant performance improvements across various datasets and configurations. Our study focuses on addressing the critical ambiguities surrounding architecture parameters within the DARTS framework, enhancing theoretical understanding and laying a robust foundation for developing more effective and versatile differentiable NAS methodologies. These advancements have the potential to extend beyond BOSE-NAS, contributing to the broader evolution of NAS research.

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

# A  APPENDIX

## A.1  DETAILED THEORETICAL PROOF OF STABLE EQUILIBRIUM STATE

The architecture updates in DARTS (Liu et al., 2018b) can be updated as:

$$\frac{d\mathcal{L}_{\text{valid}}}{d\alpha}(\alpha) = \nabla_\alpha \mathcal{L}_{\text{valid}}\left(\alpha, \omega'\right) - \frac{\xi}{2\epsilon}\left(\nabla_\alpha \mathcal{L}_{\text{train}}\left(\alpha, \omega^+\right) - \nabla_\alpha \mathcal{L}_{\text{train}}\left(\alpha, \omega^-\right)\right) \tag{14}$$

Based on Equation 14, we perform a Taylor expansion for validation loss and training loss:

$$\frac{d\mathcal{L}_{\text{valid}}}{d\alpha}(\alpha) = \frac{\Delta\mathcal{L}_{\text{valid}}\left(\alpha, \omega'\right)}{\Delta\alpha_\varepsilon} - \frac{\xi}{2\epsilon}\left(\frac{\Delta\mathcal{L}_{\text{train}}\left(\alpha, \omega^+\right)}{\Delta\alpha_\varepsilon} - \frac{\Delta\mathcal{L}_{\text{train}}\left(\alpha, \omega^-\right)}{\Delta\alpha_\varepsilon}\right) + o\left(\Delta\alpha_\varepsilon\right) \tag{15}$$

where $\xi$ is the learning rate of weight parameters $\omega$, $\epsilon$ is a small scalar related on $\omega$, and $\Delta\alpha_\varepsilon$ is the change in architecture parameter $\alpha$ between time steps $t$ and $t + \varepsilon$, where $\varepsilon$ is an infinitesimal scalar related on $t$. Among them, in the original DARTS paper, two methods for updating parameters are

proposed: first-order and second-order updates. The term $\frac{\xi}{2\epsilon}\left(\frac{\Delta\mathcal{L}_{\text{train}}\left(\alpha,\omega^+\right)}{\Delta\alpha_\varepsilon} - \frac{\Delta\mathcal{L}_{\text{train}}\left(\alpha,\omega^-\right)}{\Delta\alpha_\varepsilon}\right)$ repre-

sents an approximation of the second-order term. In this paper, we adhere to the first-order optimiza-

tion principles outlined in DARTS Algorithm 1, thus the term $\frac{\xi}{2\epsilon}\left(\frac{\Delta\mathcal{L}_{\text{train}}\left(\alpha,\omega^+\right)}{\Delta\alpha_\varepsilon} - \frac{\Delta\mathcal{L}_{\text{train}}\left(\alpha,\omega^-\right)}{\Delta\alpha_\varepsilon}\right)$

can be neglected. Consequently, Equation 15 can be approximated as:

$$\frac{\mathrm{d}\mathcal{L}_{\text{valid}}}{\mathrm{d}\alpha}(\alpha) \approx \frac{\Delta\mathcal{L}_{\text{valid}}\left(\alpha,\omega'\right)}{\Delta\alpha_\varepsilon} \tag{16}$$

Let $\alpha_t$ be the architecture parameter with $\Delta_t$ epoch updates from the initial $\alpha_0$:

$$\alpha_t = \alpha_0 - \eta_1\frac{\mathrm{d}\mathcal{L}_{\text{valid}}}{\mathrm{d}\alpha}(\alpha) = \alpha_0 - \eta_2\Delta t\frac{\mathrm{d}\mathcal{L}_{\text{valid}}}{\mathrm{d}\alpha}(\alpha) \tag{17}$$

Among them, $\eta_1$ and $\eta_2$ are the learning rate of $\alpha$, and $\eta_1 = \eta_2\Delta t$. Based on Equation 17, we now have:

$$\frac{\Delta\alpha_t}{\Delta t} = \eta_2\frac{\mathrm{d}\mathcal{L}_{\text{valid}}}{\mathrm{d}\alpha}(\alpha) \tag{18}$$

where $\Delta\alpha_t = \alpha_0 - \alpha_t$. Based on Equation 16 and 18, we have:

$$\frac{\Delta\mathcal{L}_{valid}\left(\alpha,\omega'\right)}{\Delta\alpha_\varepsilon} \approx \frac{1}{\eta_2}\frac{\Delta\alpha_t}{\Delta t} \tag{19}$$

Hence, we introduce the concept of the "Stable Equilibrium State", which provides essential insights into the validation loss trajectory across architecture spaces, as shown in Equation 20:

$$\left|\frac{\Delta s}{\Delta t}\right| \approx \left|\frac{1}{\eta_2}\frac{\Delta\rho}{\Delta t}\right| \tag{20}$$

where $\rho = \left|\frac{\Delta\alpha}{\Delta t}\right|$, $s = \left|\frac{\Delta\mathcal{L}_{\text{valid}}\left(\alpha,\omega'\right)}{\Delta\alpha_\varepsilon}\right|$.

### A.2 Detailed Theoretical evaluation of $E_\mathcal{I}$'s reliability

Consequently, we introduce Equilibrium Influential ($E_\mathcal{I}$), a novel metric designed to assess the significance of operations:

$$E_\mathcal{I} = \text{Sigmoid}\left(\frac{\Delta\alpha_{t+\Delta t} - \Delta\alpha_t}{\Delta t^2} + \frac{\Delta\alpha_t - \Delta\alpha_{t-\Delta t}}{\Delta t^2} + \frac{\Delta\alpha_{t-\Delta t} - \Delta\alpha_{t-2\Delta t}}{\Delta t^2}\right) \tag{21}$$

$$+ \frac{\alpha_{t+\Delta t} - \alpha_t}{\Delta t} + \frac{\alpha_t - \alpha_{t-\Delta t}}{\Delta t} + \frac{\alpha_{t-\Delta t} - \alpha_{t-2\Delta t}}{\Delta t}$$

By rewriting Equation 21 and conducting Taylor expansion on $\frac{\Delta\bar{\alpha}}{\Delta\bar{t}}$ for $\text{Sigmoid}\left(\frac{\Delta\alpha_{t+\Delta t} - \Delta\alpha_{t-2\Delta t}}{\Delta t^2}\right)$:

$$E_\mathcal{I} = \text{Sigmoid}\left(\frac{\Delta\alpha_{t+\Delta t} - \Delta\alpha_{t-2\Delta t}}{\Delta t^2}\right) + \frac{\alpha_{t+\Delta t} - \alpha_{t-2\Delta t}}{\Delta t} = \text{Sigmoid}\left(\eta_2\frac{\Delta\left(3\frac{\Delta\mathcal{L}_{\text{valid}}\left(\alpha,\omega'\right)}{\Delta\alpha_\varepsilon}\right)}{\Delta t}\right) + 3\frac{\Delta\bar{\alpha}}{\Delta\bar{t}} \tag{22}$$

$$= \frac{1}{1 + e^{-\frac{\Delta\bar{\alpha}}{\Delta\bar{t}}}} + \frac{e^{-\frac{\Delta\bar{\alpha}}{\Delta\bar{t}}}}{\left(1 + e^{-\frac{\Delta\bar{\alpha}}{\Delta\bar{t}}}\right)^2}\left(3\eta_2\frac{3\Delta\left(\frac{\Delta\mathcal{L}_{\text{valid}}\left(\alpha,\omega'\right)}{\Delta\alpha_\varepsilon}\right)}{\Delta\bar{t}} - \frac{\Delta\bar{\alpha}}{\Delta\bar{t}}\right) + o\left(\frac{\Delta\alpha_{t+\Delta t} - \Delta\alpha_{t-2\Delta t}}{\Delta t^2} - \frac{\Delta\bar{\alpha}}{\Delta\bar{t}}\right) + 3\frac{\Delta\bar{\alpha}}{\Delta\bar{t}}$$

where $\frac{\Delta\bar{\alpha}}{\Delta\bar{t}} = \frac{\alpha_{t+\Delta t} - \alpha_{t-2\Delta t}}{3\Delta t}$, $o\left(\frac{\Delta\alpha_{t+\Delta t} - \Delta\alpha_{t-2\Delta t}}{\Delta t^2} - \frac{\Delta\bar{\alpha}}{\Delta\bar{t}}\right)$ is the truncation error. Assuming

$\eta_2\frac{\Delta\left(\frac{\Delta\mathcal{L}_{\text{valid}}\left(\alpha,\omega'\right)}{\Delta\alpha_\varepsilon}\right)}{\Delta\bar{t}} \approx 0$ at the Stable Equilibrium State, we obtain:

$$E_\mathcal{I} \approx \frac{1}{1 + e^{-\frac{\Delta\bar{\alpha}}{\Delta\bar{t}}}} + \frac{e^{-\frac{\Delta\bar{\alpha}}{\Delta\bar{t}}}}{\left(1 + e^{-\frac{\Delta\bar{\alpha}}{\Delta\bar{t}}}\right)^2}\left(-\frac{\Delta\bar{\alpha}}{\Delta\bar{t}}\right) + \left(3\frac{\Delta\bar{\alpha}}{\Delta\bar{t}}\right) \tag{23}$$

Among them, function $f(x) = \frac{1}{1+e^{-x}} - \frac{e^{-x}}{(1+e^{-x})^2}x + 3x$ is monotonically increasing when $x$ is between $-1$ and $1$. Hence, we can obtain that $E_{\mathcal{I}}$ positively correlates with $\frac{\Delta\bar{\alpha}}{\Delta\bar{t}}$, denoted as:

$$E_{\mathcal{I}} \propto \frac{\Delta\bar{\alpha}}{\Delta\bar{t}} \tag{24}$$

Motivated by DARTS-IM (Zhang et al., 2022), which creatively introduces the influence functions to estimate the importance of operations on DARTS by estimating how the validation loss will change after posing a change on an operation. We leverage the Influence Function (Meng et al., 2020) to theoretically validate the reliability of the $E_{\mathcal{I}}$ metric as a measure of operation importance. This approach implies that the selection of operations can result in changes to the model parameters $\theta$. Therefore, the influence of candidate operations on the validation loss, denoted as $I(\theta, L)$, can be estimated as (Meng et al., 2020; Koh & Liang, 2017):

$$I(\theta, \mathcal{L}) = \frac{\mathrm{d}\mathcal{L}(\theta, \alpha)}{\mathrm{d}\epsilon} = -\nabla_{\alpha}\mathcal{L}(\theta, \alpha)^T H_{\alpha}^{-1} \nabla_{\alpha}\mathcal{L}(\theta, \alpha) \tag{25}$$

As the influence on validation loss is positively correlated with the absolute value of derivates of validation loss w.r.t (Meng et al., 2020; Koh & Liang, 2017):

$$I(\theta, \mathcal{L}(\theta, \alpha)) \propto \nabla_{\alpha}\mathcal{L}(\theta, \alpha) \tag{26}$$

Combining the Equations 19, 24, and 26, we derive that $E_{\mathcal{I}}$ positively correlates with the influence on the validation loss the magnitude of metric, denoted as:

$$E_{\mathcal{I}} \propto I(\theta, \mathcal{L}(\theta, \alpha)) \tag{27}$$

### A.3 HYPER-PARAMETERS IMPACT EVALUATION

In addition, we conduct additional experiments to analyze the impact of hyperparameters such as the learning rate and batch size on the stability and effectiveness of our method. We conducted an ablation study setting the learning rates at 0.025, 3e-3 and 1e-4, and the batch sizes at 64, 32 and 16, respectively. The performance of different learning rates in NAS-Bench-201 is shown in Table 5, while the performance of different batch sizes in NAS-Bench-201 is presented in Table 6.

We observe that, although there are slight variations in performance due to different hyperparameters, our method consistently identifies architectures with superior performance. This highlights the generality and robustness of BOSE-NAS.

### A.4 PERFORMANCE EVALUATION IN TRANSFORMER-BASED SEARCH SPACE

To verify the generalization and robustness of BOSE-NAS, we applied it to optimize the fine-tuning process of ALBERT (Lan et al., 2019), a large pre-trained Transformer-based model. Fine-tuning large pre-trained models is critical for transfer learning in various scenarios. However, this approach often suffers from parameter inefficiency when addressing multiple downstream tasks, as each task requires a separate model. Adapter (Houlsby et al., 2019) modules offer a more efficient alternative, introducing a small number of trainable parameters for each task while preserving scalability. The architecture of the adapter significantly impacts both performance and parameter efficiency. However, manually selecting the optimal architecture is resource intensive and often suboptimal.

To address this, we utilize BOSE-NAS to automate the search for adapter architectures, balancing accuracy and computational efficiency. The search space is defined as: {Identity Mapping, Self-Attention Layer, 1D-Convolutional Layer (Conv1 $\times$ 1), Multi-Layer Perceptron (MLP)}

The experimental results, summarized in Table 7, demonstrate that BOSE-NAS efficiently identified the optimal adapter architecture, achieving greater precision with fewer fine-tuned parameters compared to traditional full fine-tuning approaches. These findings highlight the effectiveness of BOSE-NAS in balancing performance and efficiency, making it a valuable tool for improving fine-tuning processes in Transformer-based models.

### A.5 APPLICATION IN REAL-WORLD SCENARIOS

To further validate the generalization ability, robustness, and potential applications of our method, we applied it to real-world image classification and text recognition tasks. The first task is to classify

Table 5: The performance of different learning rates on CIFAR-10, CIFAR-100, and ImageNet datasets in NAS-Bench-201.

| Learning rate | Acc.(%)on CIFAR-10 | Acc.(%)on CIFAR-100 | Acc.(%)on ImageNet-16 |
|---|---|---|---|
| 0.025 | 94.37 | 73.09 | 46.63 |
| 3e-3 | 94.08 | 72.01 | 45.62 |
| 1e-4 | 94.02 | 73.00 | 45.44 |

Table 6: The performance of different batch size on CIFAR-10, CIFAR-100, and ImageNet datasets in NAS-Bench-201.

| Batch Size | Acc.(%)on CIFAR-10 | Acc.(%)on CIFAR-100 | Acc.(%)on ImageNet-16 |
|---|---|---|---|
| 64 | 94.37 | 73.09 | 46.63 |
| 32 | 94.24 | 72.76 | 46.23 |
| 16 | 94.36 | 73.51 | 46.34 |

Table 7: Accuracy and the number of parameters for different fine-tuning methods on ALBERT backbone.

| Fine-tuning methods | Acc.(%)on QNLI | Finetuned Params |
|---|---|---|
| Full-finetuning | 86.27 | 11,683,584 |
| Adapter | 86.49 | 617,856 |
| Adapter+BoseNAS | 87.01 | 631,296 |

Table 8: Result of the store classification task.

| | test accuracy (%) | param (M) |
|---|---|---|
| ResNet-50 | 57.04 | 23.55 |
| ResNet-101 | 59.24 | 42.54 |
| MobileNet-v3-small | 42.2 | 1.25 |
| MobileNet-v3-large | 50.68 | 2.7 |
| **BOSE-NAS** | 59.24 | 4.26 |

Table 9: Result of the business license recognition task.

| | test accuracy (%) | param (M) |
|---|---|---|
| ResNet-aster | 95.02 | 15.5 |
| MobileNet-v3-small | 95.22 | 3.74 |
| **BOSE-NAS** | 95.47 | 3.87 |

Table 10: Results for DARTS+ and BOSE-NAS on CIFAR-10, CIFAR-100, and ImageNet datasets in NAS-Bench-201.

|                      | Acc.(%)on CIFAR-10 | Acc.(%)on CIFAR-100 | Acc.(%)on ImageNet-16 |
|----------------------|--------------------|---------------------|-----------------------|
| DARTS+(Criterion 1)  | $92.50 \pm 0.06$   | $69.11 \pm 0.14$    | $42.09 \pm 0.00$      |
| DARTS+(Criterion 2)  | $90.59 \pm 0.00$   | $67.34 \pm 0.00$    | $40.08 \pm 0.00$      |
| **BOSE-NAS**         | $94.21 \pm 0.22$   | $72.72 \pm 0.51$    | $46.27 \pm 0.08$      |

the category of the store, consisting of 76,189 images in 21 categories, divided into a training set of 29,159 images, a validation set of 23,512 images and a test set of 23,518 images, all in 3 channel RGB format. The second task is for business license content recognition, in which the dataset includes approximately 1.46 million images of business licenses splitting into training and test sets at an 8:2 ratio, also in 3-channel RGB format.

As shown in Table 8, our method achieved a test accuracy of $59.24\%$ for store classification, ranking first among the compared methods while being ten times more parameter-efficient than ResNet-101. For the content recognition task, based on the CRNN framework, as shown in Table 9, our method reached a test accuracy of $95.47\%$, surpassing competing methods with more than four times the parameter efficiency compared to the ResNet-aster model. These results demonstrate the effectiveness of our method in real-world applications.

### A.6 COMPARISON WITH DARTS+, DARTS- AND $\beta$-DARTS

DARTS+ (Liang et al., 2019) attributes the collapse issue to overfitting during the optimization process in DARTS. To address this, it introduces two early stopping criteria: one that halts the search when the ranking of architecture parameters for learnable operations stabilizes over a specified number of epochs and another that stops the process when two or more skip connections appear in a normal cell. Rather than relying on a heuristic early stopping mechanism, we introduce the concept of a Stable Equilibrium State, grounded in rigorous theoretical analysis, to represent the stability of supernet training. By tracking this state throughout the training process, we determine the optimal point to stop training and begin architecture derivation upon encountering the first minima. Importantly, in our approach, this minima could occur at any stage of the training process.

To further illustrate the differences between our method and DARTS+, we conducted an empirical analysis using the NAS-Bench-201 and DARTS search spaces, with results shown in Table 10 and Table 11. In NAS-Bench-201, BOSE-NAS outperforms DARTS+ by a significant margin. In the DARTS search space, BOSE-NAS achieves an average test accuracy of $97.51\%$ on CIFAR-10, slightly surpassing DARTS+, while being more than three times as efficient. On CIFAR-100, BOSE-NAS attains an average test accuracy of $83.77\%$, which also outperforms DARTS+. These experiments demonstrate that our method offers superior accuracy, search efficiency, and generalizability compared to DARTS+, thereby highlighting the advantages of our approach.

In addition, previous studies, such as DARTS- (Chu et al., 2020a) and $\beta$-DARTS (Ye et al., 2022) have successfully controlled architecture parameters to achieve robust results. Specifically, DARTS- introduces an auxiliary skip connection to ensure fair competition among operations, thereby controlling the updates of architecture parameters. Similarly, $\beta$-DARTS employs $\beta$-Decay regularization to maintain the stability and variance of activated parameters. In our approach, while we utilize architecture parameters, we do not directly control them. Instead, we first theoretically demonstrate that the change rate of architecture parameters signifies the sensitivity of the validation loss. Leveraging this insight, we develop a metric to track the supernet's training trajectory. Furthermore, we design new operation importance measurements based on the first and second-order differentials of the architecture parameters.

From another perspective, DARTS- and $\beta$-DARTS aim to enhance the correlation between architecture parameters and the importance of operations by controlling or modifying the updates of alpha. In contrast, our method maps alpha to a different space and devises a new metric, facilitating a more effective assessment of operation importance.

Table 11: Results for DARTS+ and BOSE-NAS on CIFAR-10 and CIFAR-100 datasets in DARTS search spaces.

|  | Test Err.(%)on CIFAR-10 | Test Err.(%)on CIFAR-100 | Search Cost (GPU-days) |
|---|---|---|---|
| DARTS+ | $2.50 \pm 0.11$ | 16.28 | 0.4 |
| **BOSE-NAS** | $2.49 \pm 0.11$ | $16.23 \pm 0.11$ | 0.13 |

## A.7 ALGORITHM

---

**Algorithm 1** BOSE-NAS

---

**Require:** Create a mixed operation $\bar{o}(i, j)$ parameterized by $\alpha(i, j)$ for each edge $(i, j)$, set training epochs $N_1$, set epoch thresh $N_2$.

    **for** training epoch $n$ in $N_1$ **do**

        update architecture $\alpha$ by descending $\nabla_\alpha \mathcal{L}_{\text{valid}} (\omega - \xi \nabla_\omega \mathcal{L}_{\text{train}} (\omega, \alpha), \alpha)$.

        update weights $\omega$ by descending $\nabla_\omega \mathcal{L}_{\text{train}} (\omega, \alpha)$.

        **if** $n > N_2$ **then**

            Calculate the Stable Equilibrium State by Equation 9.

            **if** reach optimal Stable Equilibrium State **then**

                calculate $E_{\mathcal{I}}$ by Equation 10.

                derives the final architecture based on the $E_{\mathcal{I}}$.

                **return**

            **end if**

        **end if**

    **end for**

---

