# OpenReview forum: "BOSE-NAS: Differentiable Neural Architecture Search with Bi-Level Optimization Stable Equilibrium"
_ICLR.cc/2025/Conference — Submitted to ICLR 2025_

### Official Review · Reviewer_B6J3 · 2024-10-23

**Soundness:** 2
**Presentation:** 2
**Contribution:** 2
**Rating:** 3
**Confidence:** 4

**Summary:**

This paper proposes a new operation importance evaluation metric in network architecture search. The authors first introduce the concept of stable equilibrium state, which shows the stability of the bi-level optimization process in differentiable NAS. By analyzing the supernet training dynamics, the metric named equilibrium influential is proposed for fair differentiable NAS. The experimental results show that the proposed metric and search method can achieve competitive accuracy with significantly reduced search cost.

**Strengths:**

+ The experimental results clearly show the effectiveness and the efficiency of the proposed method.

**Weaknesses:**

- The writing can be improved. The abstract and the introduction are redundant. For the abstract, there are too many contents to introduce the background. For the introduction, many details especially the experimental results don’t have to be elaborated. I think demonstrating the main results is enough to show the effectiveness of this method.

- The technical soundness can be further verified. There are some strong assumptions without verification or explanation. For example, the assumptions to transit (6) to (7) should be verified. Why they have little effect on $\alpha$?

- Some exact calculations can be put in the Appendix part.

- The reason why the proposed method has less search cost should be analyzed in the result analysis, which is an important benefit from the new metric.

- The performance of the proposed method underperforms the SOTA NAS methods such as IS-DARTS. More clarification is required for the performance analysis.

**Questions:**

See Weakness.

---

> ### Author Response · Authors · 2024-11-15
> **Response to weakness 1-3**
>
> We sincerely thank the reviewers for their detailed feedback and constructive suggestions. We have carefully considered each comment and provide our point-by-point responses below.
>
> **Responses to weakness 1:**
>
> Thank you for highlighting this concern. We agree that there is redundancy in the abstract and introduction. We have revised the Abstract section to focus more on the main contributions and key results, minimizing background information and condense the Introduction by reducing detailed descriptions of experimental results, emphasizing the main findings to better highlight the effectiveness of the proposed method. And we also present the revised section below:
>
> Abstract is revised as: Recent research has significantly mitigated the performance collapse issue in Differentiable Architecture Search (DARTS) by either refining architecture parameters to better reflect the true strengths of operations or developing alternative metrics for evaluating operation significance. However, the actual role and impact of architecture parameters remain insufficiently explored, creating critical ambiguities in the search process. To address this gap, we conduct a rigorous theoretical analysis demonstrating that the change rate of architecture parameters reflects the sensitivity of the supernet’s validation loss in architecture space, thereby influencing the derived architecture's performance by shaping supernet training dynamics. Building on the these insights, we introduce the concept of a Stable Equilibrium State to capture the stability of the bi-level optimization process and propose the Equilibrium Influential ($E_\mathcal{I}$) metric to assess operation importance. By integrating these elements, we propose BOSE-NAS, a differentiable NAS approach that leverages the Stable Equilibrium State to identify the optimal state during the search process and derive the final architecture using the $E_\mathcal{I}$ metric. Extensive experiments across diverse datasets and search spaces demonstrate that BOSE-NAS achieves competitive test accuracy compared to state-of-the-art methods while significantly reducing search costs.
>
> Introduction (line 76-85) is revised as：In the DARTS search space, BOSE-NAS achieves an impressive average test error of 2.49% and a best test error of 2.37% on the CIFAR-10 dataset. When transferred to CIFAR-100 and ImageNet, BOSE-NAS attains an average test error of 16.23% and a best test error of 16.08% on CIFAR-100, and a best test error of 24.1% on ImageNet. Remarkably, our method accomplishes this with a mere 0.13 GPU-days of computational cost (equivalent to just 3 hours of search time on a single V100 GPU) for architecture search on CIFAR-10. This level of efficiency outperforms DARTS by more than three times and surpasses DARTS-PT by nearly six times.
>
> **Responses to weakness 2:**
>
> In the original DARTS paper, two methods for updating parameters are proposed: first-order and second-order updates. The term $\frac{\xi}{2 \epsilon}(\frac{\Delta L_{train}(\alpha, \omega^{+} )}{\Delta\alpha_{\varepsilon}}-\frac{\Delta L_{train}(\alpha, \omega^{-} )}{\Delta\alpha_{\varepsilon}})$ presented in Equation 6 represents an approximation of the second-order term. In this paper, we adhere to the first-order optimization principles outlined in DARTS Algorithm 1, thus the term $\frac{\xi}{2 \epsilon}(\frac{\Delta L_{train}(\alpha, \omega^{+} )}{\Delta\alpha_{\varepsilon}}-\frac{\Delta L_{train}(\alpha, \omega^{-} )}{\Delta\alpha_{\varepsilon}})$ can be disregarded.
>
> In addition, we will incorporate a detailed, step-by-step theoretical proof and explanation in the Appendix of the revised manuscript.
>
> **Responses to weakness 3:**
>
> We appreciate this suggestion for better structuring our content. We will include the detailed theoretical calculations and derivations to the Appendix in the revised version.

---

> ### Author Response · Authors · 2024-11-15
> **Responses to weakness 4-5**
>
> **Responses to weakness 4:**
>
> Thank you for bringing up this valuable point. We agree that a more comprehensive analysis of the search cost efficiency would strengthen the discussion in our results section. In the revised manuscript, we will elaborate on the key factors contributing to the superior search efficiency of BOSE-NAS, which are outlined as follows:
>
> Early Identification of a Stable Supernet: BOSE-NAS demonstrates its search efficiency primarily through its capability to identify a stable state at an early stage of the supernet training process. This early identification enables the derivation of the final architecture without the need for prolonged training phases. This approach aligns with findings from related research[1,2,3] that shows extended training of the supernet is often unnecessary for accurately assessing the relative strength among operations, thus reducing overall search cost.
>
> Efficient Evaluation Metric: Unlike other methods that may involve higher computational complexity for assessing operation strength, such as those used in [3-6], the proposed EI metric operates with lower overhead while maintaining reliable performance. This streamlined process contributes significantly to the reduced search cost and ensures rapid and consistent evaluations.
>
> We will include a detailed explanation of this analysis in the revised result section to better highlight the efficiency benefits provided by BOSE-NAS.
>
> [1] Hanwen Liang, Shifeng Zhang, Jiacheng Sun, Xingqiu He, Weiran Huang, Kechen Zhuang, and Zhenguo Li. Darts+: Improved differentiable architecture search with early stopping. arXiv preprint arXiv:1909.06035, 2019.
>
> [2] ZELA A, ELSKEN T, SAIKIA T, et al. Understanding and Robustifying Differentiable Architecture Search[J]. International Conference on Learning Representations,International Conference on Learning Representations, 2020.
>
> [3] Xin Chen, Lingxi Xie, Jun Wu, and Qi Tian. Progressive darts: Bridging the optimization gap for
> nas in the wild. International Journal of Computer Vision, 129:638–655, 2021b.
>
> [4] HONG W, LI G, ZHANG W, et al. DropNAS: Grouped Operation Dropout for Differentiable Architecture Search[C/OL]//Proceedings of the Twenty-Ninth International Joint Conference on Artificial Intelligence, Yokohama, Japan. 2020.
>
> [5] WANG R, CHENG M, CHEN X, et al. Rethinking Architecture Selection in Differentiable NAS[J]. International Conference on Learning Representations,International Conference on Learning Representations, 2021.
>
> [6] Li, G., Zhang, X., Wang, Z., Li, Z., Zhang, T.: StacNAS: Towards stable and consistent optimization for differentiable Neural Architecture Search. arXiv preprint arXiv:1909.11926 (2019)
>
> **Responses to weakness 5:**
>
> We understand the concern regarding the performance comparison with SOTA NAS methods.
>
> While it is true that our method achieves performance comparable to existing state-of-the-art (SOTA) methods in some of the experiments, we would like to emphasize that the primary contribution of our research extends beyond empirical accuracy.
>
> The central objective of our study is to address and resolve the ambiguities surrounding the actual role and impact of architecture parameters within the DARTS framework. This focus is critical for enhancing the theoretical understanding and robustness of differentiable NAS methods. We believe that filling these gaps and proposing a more analytically grounded differentiable NAS approach contributes significant value to the field, complementing empirical findings with deeper scientific insights. This combination of practical results and theoretical advancement broadens the understanding and potential applications of differentiable NAS methodologies.
>
> We hope this response clarifies our perspective and underscores the importance of our contributions beyond raw performance metrics.

---

> ### Author Response · Authors · 2024-11-22
> **We have uploaded the latest version of the manuscript**
>
> Thank you once again for your invaluable suggestions and the time you have invested in reviewing our manuscript. We have now uploaded the revised version of the manuscript. All changes have been meticulously integrated into the latest version, and they are highlighted for your convenience during the review process. We sincerely hope that these enhancements align with your expectations.
>
> To elaborate, based on the reviewers' recommendations, we have added a comprehensive, step-by-step theoretical proof and explanation in the Appendix of the revised manuscript. Additionally, we have expanded our discussion on the critical elements that contribute to the superior search efficiency of BOSE-NAS. The Abstract has been refined to emphasize the primary contributions and key findings, while reducing background information. We have also condensed the Introduction by streamlining detailed descriptions of experimental results, thereby placing greater emphasis on the main findings to better showcase the effectiveness of our proposed method.
>
> Having thoroughly addressed the questions raised, we kindly and respectfully invite you to consider revisiting your initial rating. Should you not foresee adjusting your rating, we would greatly appreciate any further insight you could offer regarding whether this stance stems from lingering concerns about specific experimental aspects or reflects reservations about the broader direction of our research. We are very much looking forward to maintaining an ongoing dialogue with you and remain profoundly thankful for the meticulous attention and dedication you have shown in evaluating our submission. Your feedback is immensely valuable to us.

---

> > ### Comment · Reviewer_B6J3 · 2024-11-27
> >
> > Thank you for the response. The performance of the proposed method is still the major concern, so I will keep my score.

---

> > > ### Author Response · Authors · 2024-11-29
> > > **Response to your concerns**
> > >
> > > We appreciate the opportunity to address your concerns regarding the empirical performance of our method. Due to time and resource constraints during our initial experiments, our primary focus was on validating the theoretical contributions rather than optimizing empirical accuracies.
> > >
> > > In our initial experiments:
> > >
> > > ● DARTS and S1-S4 Search Spaces: Our method consistently outperformed or matched state-of-the-art (SOTA) methods.
> > >
> > > ● NAS-Bench-201: Our results were comparable to SOTA methods.
> > >
> > > In response to your concern, we have conducted additional experiments to further validate the empirical performance of our method on the NAS-Bench-201 search space. The updated results, presented in Table 1, demonstrate improvements that further affirm the competitiveness of our approach:
> > >
> > > Table 1 Comparison with state-of-the-art method on NAS-Bench-201.
> > > | Method       | CIFAR-10 | CIFAR-100 | ImageNet-16-120 |
> > > |-----------------|---------------|----------------|----------------------|
> > > | ResNet (2016)        | 93.97         | 70.86          | 43.63                |
> > > | Random                          | 93.70±0.36    | 70.65±1.38     | 90.93±2.15           |
> > > | ENAS   | 54.30±0.00    | 10.62±0.27     | 16.32±0.00           |
> > > | DARTS(2018)  | 54.30±0.00    | 15.61±0.00     | 16.32±0.00           |
> > > | SNAS(2018)  | 92.77±0.83    | 69.34±1.98     | 43.16±2.64           |
> > > | GDAS(2019) | 93.23±0.23    | 24.20±8.08     | 41.02±0.00           |
> > > | PC-DARTS(2019)| 93.41±0.30    | 67.48±0.89     | 41.31±0.22           |
> > > | DrNAS(2020)| 94.36±0.00    | 73.51±0.00     | 46.34±0.00           |
> > > | IDARTS(2021) | 93.58±0.32 | 70.83±0.48 | 40.89±0.68           |
> > > | DARTS-PT(2021) | - | 93.80 | -                   |
> > > | DARTS-PT(fix alpha)| - | 93.61±0.23 | -                   |
> > > | DARTS-IM(2022)| 94.36±0.00 | 73.51±0.00 | 46.34±0.00           |
> > > | β-DARTS(2022) | 94.36±0.00    | 73.51±0.00     | 46.34±0.00           |
> > > | OLES(2023) | 93.70±0.15    | 70.40±0.22     | 43.97±0.38           |
> > > | IS-DARTS(2024) | 94.36±0.00    | 73.51±0.00     | 46.34±0.00           |
> > > | **BOSE-NAS (Avg)**  | **94.37±0.01**    | 73.37±0.24     | **46.34±0.01**   |
> > > | **BOSE-NAS (Best)**  | **94.37**         | **73.51**          | **46.34**                |
> > >
> > > While we respect your perspective, we would like to emphasize that the primary contribution of our research extends beyond raw performance metrics. The central objective of our study is to address the ambiguities surrounding the role and impact of architecture parameters within the DARTS framework. By providing a deeper theoretical understanding of these parameters, our work aims to facilitate more effective utilization of these parameters, laying the groundwork for the development of advanced differentiable NAS methods and broadening their potential applications.
> > >
> > > We hope these additional experiments, coupled with our focus on addressing foundational challenges, provide further evidence of the significance and potential of our work. Thank you again for your valuable feedback.

---

### Official Review · Reviewer_b8cC · 2024-10-24

**Soundness:** 3
**Presentation:** 2
**Contribution:** 2
**Rating:** 5
**Confidence:** 4

**Summary:**

Differentiable Architecture Search (DAS) often faces the issue where the magnitude of architecture parameters fails to reflect the true importance of operations. This paper addresses this problem by proposing BOSE-NAS, a DAS method guided by the Stable Equilibrium of architecture parameters (i.e., the point where the rate of change of the architecture parameters is minimal). The authors provide relevant experiments to support their method. However, the experimental section has several issues, such as limited improvement in performance and a lack of ablation studies.

**Strengths:**

1.	The paper is easy to read.
2.	The problem of DAS is clear.

**Weaknesses:**

This paper was submitted to NeurIPS 2024, compared with NeurIPS 2024, there are still some important issues that need to be addressed.
1. The ablation studies are not convincing. To be specific, in Figure 3, we can clearly see that the proposed method is sensitive to hyperparameters.
2. There still exist some typos/grammatical errors in the paper.
3. The format of references is still wrong.
4. Exploring the reasons behind the success of these techniques and providing intuitive explanations would contribute to the overall scientific contribution of the work.
5. I don't understand the theoretical analysis. Why use " Influence Function"? What relationship between " Influence Function" and your method? why validate the "reliability" of your proposed metric? Please provide detailed motivation and clear proven process in step by step. What is the difference between stability and reliability? Please provide a step-by-step proof process for validating their metric. And, clarification on the relationship between the Influence Function and their method.
6. In page 7, "I(z, L)" denotes?
7. The main limitation of this paper is that proposed method lacks comparison with larger datasets (i.e., COCO2017, VOC), and more competitors (i.e., β-DARTS++, Λ-DARTS).
8. Pls to prove your statement of generalizability.

[1] β-DARTS++: Bi-level Regularization for Proxy-robust Differentiable Architecture Search
[2] Λ-DARTS: MITIGATING PERFORMANCE COLLAPSE BY HARMONIZING OPERATION SELECTION AMONG CELLS

**Questions:**

pls see weaknesses

---

> ### Author Response · Authors · 2024-11-15
> **Responses to weakness 1-4**
>
> We would like to express our gratitude to the reviewers for their thoughtful and detailed feedback on our manuscript.
>
> **Response to weakness1:**
>
>  Thank you for your observation. Since multiple local Stable Equilibrium State minima may exist with extended supernet training as shown in Fig.2, the primary aim of the experiment depicted in Figure 3 is to support our strategy of designating the first Stable Equilibrium State as the optimal point for architecture derivation. This choice is a design decision and is largely independent of the hyperparameters of our method.
>
> **Response to weakness2:**
>
>  We apologize for any oversight regarding typos or grammatical issues in the initial submission. We have meticulously reviewed the entire manuscript and corrected all identified errors to enhance readability and coherence. And all revisions will be updated in the revised version of the article.
>
> **Response to weakness3:**
>
> We sincerely apologize for the oversight in the formatting of our references. After receiving your valuable feedback, we identified that some inconsistencies were caused by issues with our reference management software, which led to missing or incorrectly formatted information. We have meticulously reviewed the entire Reference section and have corrected all formatting inconsistencies to fully comply with the journal’s guidelines. This revision will be included in the updated manuscript and thank you for bringing this to our attention.
>
> **Response to weakness4:**
>
> We appreciate your suggestion. Below, we outline the reasons and provide intuitive explanations for the success of our approach:
>
> The core objective of our work is to resolve ambiguities surrounding the actual role and impact of architecture parameters in DARTS, facilitating the development of more effective differentiable NAS methodologies. Our theoretical analysis reveals that the change rate of architecture parameters reflects the sensitivity of the supernet’s validation loss, which shapes the training dynamics of the supernet, ultimately influencing the performance of the derived architecture.
>
> Empirical studies have shown that while prolonged supernet training may lead to overfitting and degrade the final architecture’s performance [1-3], a supernet experiencing significant fluctuations during training can also result in poor final performance [2, 4]. Thus, identifying a stable state helps prevent these issues and improves the architecture derivation process.
>
> Building on our findings, the Stable Equilibrium State metric is presented to effectively track the trajectory of validation loss across the architecture space. It allows us to monitor and identify a supernet with a stable state, providing an optimal point for architecture derivation.
>
> The Equilibrium Influential (EI) metric we proposed plays a crucial role in assessing the relative strength among operations in the supernet. Intuitively, the proposed EI metric assesses the influence of operations on the stability of the supernet and quantifies their contribution to maintaining a stable state. Our theoretical analysis supports that the magnitude of the EI metric is positively correlated with an operation’s influence on the validation loss, establishing it a reliable measure for determining the relative importance of operations in a stable supernet.
>
> Together, these techniques form the foundation of BOSE-NAS, which has been demonstrated to effectively identify competitive architectures across diverse search spaces and datasets.
>
> [1] Hanwen Liang, Shifeng Zhang, Jiacheng Sun, Xingqiu He, Weiran Huang, Kechen Zhuang, and Zhenguo Li. Darts+: Improved differentiable architecture search with early stopping. arXiv preprint arXiv:1909.06035, 2019.
>
> [2] ZELA A, ELSKEN T, SAIKIA T, et al. Understanding and Robustifying Differentiable Architecture Search[J]. International Conference on Learning Representations,International Conference on Learning Representations, 2020.
>
> [3] Xin Chen, Lingxi Xie, Jun Wu, and Qi Tian. Progressive darts: Bridging the optimization gap for
> nas in the wild. International Journal of Computer Vision, 129:638–655, 2021b.
>
> [4] CHEN X, HSIEH C J. Stabilizing Differentiable Architecture Search via Perturbation-based Regularization[J]. Cornell University - arXiv,Cornell University - arXiv, 2020.

---

> ### Author Response · Authors · 2024-11-15
> **Responses to weakness 5-6**
>
> **Response to weakness 5:**
>
> Thank you for this comprehensive set of questions.
>
> Motivation for Using Influence Function: The Influence Function [1][2]  is a well-established tool from robust statistics that quantifies the effect of perturbing or upweighting a specific training sample on model parameters. It has been successfully applied in various machine learning applications to explain model behavior. Different from previous works that analyzed the effects of removing data points on model parameters, DARTS-IM [3] creatively adapted the Influence Function to estimate the significance of candidate operations within a trained supernet, providing insights into operation selection in differentiable NAS methods
>
> Relationship Between Influence Function and Our Method: Motivated by [3], we leverage the concept of the Influence Function to validate the reliability of our proposed Equilibrium Influential (EI) metric. By adapting the Influence Function, we can analyze how changes in specific operations affect the validation loss. Our analysis demonstrates that the magnitude of the EI metric is positively correlated with the operation’s influence on validation loss, thus confirming its reliability as a measure of operation importance. This correlation ensures that the EI metric could reliably determine the relative significance among operations necessary for robust architecture derivation.
>
> Why Validate the Reliability of the Metric: Validating the reliability of the EI metric is crucial, as existing studies have shown that metrics failing to represent the true strength of operations can lead to degraded architectures. Ensuring our EI metric's reliability helps address the performance degradation seen in prior differentiable NAS methods and supports the robustness of BOSE-NAS.
>
> We will include a step-by-step theoretical proof process and explanation in the Appendix of the revised manuscript.
>
> [1] F. R. Hampel. The influence curve and its role in robust estimation. Journal of the american statistical association, 69(346):383–393, 1974.
>
> [2] P. W. Koh and P. Liang. Understanding black-box predictions via influence functions. In International Conference on Machine Learning, pages 1885–1894. PMLR, 2017.
>
> [3] MiaoZhang, Wei Huang, and BinYang. Interpreting operation selection in differentiable architecturesearch: A perspective from influence-directed explanations. Advances in Neural Information Processing Systems, 35: 31902–31914, 2022.
>
> **Response to weakness 6:**
>
> We thank the reviewer for highlighting this issue, and we appreciate the opportunity to clarify and correct it. In the original work [1][2], "I(z, L)" represents the effect of removing training data points on the validation loss. However, in our work, motivated by DARTS-IM [3], we adapt influence functions to estimate the significance of candidate operations within the differentiable architecture search context. Thus, in this context, it should be denoted as "I(θ, L)" instead of "I(z, L)", where "I(θ, L)" specifically represents the influence of candidate operations on the validation loss.
>
> We sincerely apologize for this oversight and assure you that the necessary corrections will be made in the final version of the manuscript. Thank you again for pointing this out.
>
> [1] F. R. Hampel. The influence curve and its role in robust estimation. Journal of the american statistical association, 69(346):383–393, 1974.
>
> [2] P. W. Koh and P. Liang. Understanding black-box predictions via influence functions. In International Conference on Machine Learning, pages 1885–1894. PMLR, 2017.
>
> [3] MiaoZhang, Wei Huang, and BinYang. Interpreting operation selection in differentiable architecturesearch: A perspective from influence-directed explanations. Advances in Neural Information Processing Systems, 35: 31902–31914, 2022.

---

> ### Author Response · Authors · 2024-11-15
> **Responses to weakness 7-8**
>
> **Responses to weakness 7:**
>
> Thank you for your valuable suggestions. The idea you proposed to evaluate our method on detection datasets, such as COCO2017 and VOC, is highly appreciated. We fully agree that extending our approach to these datasets would provide additional insights into its scalability, transferability, and generalization capabilities.
>
> However, it is important to note that most existing Neural Architecture Search (NAS) methods, including the two works you highlighted (β-DARTS++ and Λ-DARTS), have predominantly been evaluated on classification datasets. This approach aligns with the standard practice in the field, where classification tasks are widely used as foundational benchmarks for assessing the performance of NAS methods. In this context, our study follows the same tradition by validating our method on large-scale classification datasets, including ImageNet, which is a well-established benchmark and provides a rigorous and credible evaluation of our method's effectiveness and competitiveness.
>
> While incorporating comparisons on detection datasets such as COCO2017 and VOC would undoubtedly enhance the breadth of our study, such an extension was beyond the scope of the current work. Nevertheless, we view this as an important avenue for future research and will actively consider it in subsequent studies to build upon the foundation established in this paper.
>
> **Responses to weakness 8:**
>
>  We validate the generalizability of our method through extensive experiments across three different datasets and six diverse search spaces. Specifically, we show that architectures discovered using our method on CIFAR-10 successfully transfer to CIFAR-100 and ImageNet in the DARTS search space, NAS-Bench-201 search space and S1-S4 search space, maintaining competitive performance. These results affirm the strong generalization capability of our method.
>
> To further substantiate the generalization capability, robustness, and practical applicability of our proposed method, we applied it to two real-world tasks: store classification and recognition tasks, using dedicated business datasets. Detailed descriptions of these datasets can be found in the appendix of our paper. The first dataset consists of 76,189 images across 21 categories for store classification, whereas the second dataset includes approximately 1.46 million images of business licenses for text recognition. Our method not only surpassed existing approaches in performance but also achieved this with over four times greater parameter efficiency compared to the ResNet model. These results highlight the outstanding generalizability and effectiveness of our method across a variety of real-world applications.

---

> ### Author Response · Authors · 2024-11-22
> **We have uploaded the latest version of the manuscript**
>
> Thank you once again for your invaluable suggestions and questions. Based on your feedback, we have detailed the rationale behind our approach and provided clear, intuitive explanations for its success. Furthermore, we have added a comprehensive step-by-step theoretical proof and explanation in the Appendix of the revised manuscript, elucidating the connection between the Influence Function and the methods employed.
>
> We have also conducted a thorough review of the document, correcting any typographical errors, grammatical issues, and ensuring the consistency of reference formatting.
>
> The revised version of the manuscript is now available for review. All revisions have been meticulously incorporated into the latest version, with changes highlighted for your convenience.
>
> We are deeply grateful for your meticulous and thorough review. We trust that these amendments clarify our findings and ensure that previous oversights do not detract from your assessment of the manuscript. If you have any further questions or additional feedback, please feel free to reach out to us. We look forward to continuing our dialogue with you.

---

> > ### Comment · Reviewer_b8cC · 2024-11-26
> >
> > Thanks for your reply. After reading other reviews and the rebuttal, I will keep my score.

---

### Official Review · Reviewer_uGSJ · 2024-10-27

**Soundness:** 3
**Presentation:** 3
**Contribution:** 2
**Rating:** 5
**Confidence:** 5

**Summary:**

In this paper, the authors propose BOSE-NAS, a novel differentiable neural architecture search method that addresses critical challenges in existing differentiable architecture search (DARTS). The core idea of BOSE-NAS  is around the the concept of a ‘Stable Equilibrium State’, which offering insights into the validation loss trajectory across architectural spaces to stabilise the supernet’s bi-level optimisation process. The proposed method introduces a novel metric called Equilibrium Influential (EI) to evaluate the importance of operations during the architecture search phase. By choosing operations based on the EI metric at the Stable Equilibrium State, BOSE-NAS uses bi-level optimisation to find the optimal architecture operations.

**Strengths:**

1. The introduction of Stable Equilibrium State is somewhat novel and interesting, the theoretical analysis of architecture parameter dynamics provides a solid foundation for understanding the bi-level optimisation in differentiable NAS.

2. The Equilibrium Influential (EI) metric for operation evaluation is an innovative approach and offers a more reliable measure of operation importance to the bi-level optimisation process in the differentiable NAS.

2. The proposed BOSE-NAS achieves competitive performance as well as less computational overhead in benchmark datasets like CIFAR-10 and CIFAR-100, compare with other differentiable NAS methods.

**Weaknesses:**

1. The propose method heavily depends on the accurate identification of the Stable Equilibrium State, specifically, the EI metric evaluates each operation independently, which could overlook potential dependencies among network operations within the architecture. This could make the proposed method not always generalise well.

2. The biggest concern of the proposed method, e.g., EI metric and the concept of Stable Equilibrium State, are the limited use scenario. It may not be easily applicable to non differentiable NAS methods, e.g., the evolutionary or pruning-based search algorithms.

**Questions:**

1. Although the problems within the bi-level optimisation process of differentiable NAS have been widely studied for years, e.g., BONAS [1], the proposed EI metric and Stable Equilibrium State still bringing some new insights to the NAS research. But differentiable NAS are often sensitive to the hyper-parameters, I wonder how sensitive is the Stable Equilibrium State identification process to the choice of hyper-parameters such as the learning rate and batch size? Can authors provide some ablation studies? It would be helpful to understand how the proposed method handles changes in the hyper-parameters, as well as its robustness.

2. The proposed methods are only applied to the differentiable NAS, however, the interest of NAS research has been largely shifted to training-free NAS methods, as they are offering more flexibilities to different search algorithms and search spaces, as well as better performance and much less computational overhead compare with differentiable NAS, e.g., Zen-NAS [2] and SWAP-NAS [3]. Can author discuss the potential adaptation that extend the concept the Stable Equilibrium State and EI metric to non-differentiable NAS methods?



[1] Han Shi, Renjie Pi, Hang Xu, Zhenguo Li, James T. Kwok, and Tong Zhang. Bridging the gap between sample-based and one-shot neural architecture search with bonas. NeurIPS 2020.

[2] Ming Lin, Pichao Wang, Zhenhong Sun, Hesen Chen, Xiuyu Sun, Qi Qian, Hao Li, and Rong Jin. Zen-nas: A zero-shot NAS for high-performance image recognition. ICCV 2021.

[3] Yameng Peng, Andy Song, Haytham. M. Fayek, Vic Ciesielski and Xiaojun Chang . SWAP-NAS: Sample-Wise Activation Patterns for Ultra-fast NAS. ICLR 2024.

---

> ### Author Response · Authors · 2024-11-15
> **Response to weaknesses and questions.**
>
> We would like to express our sincere gratitude to the reviewers for their valuable feedback and constructive comments on our manuscript. We have carefully considered each point and provide our detailed responses below.
>
> **Response to weakness1:**
>
> Thank you for your insightful observation. Indeed, the identification of a Stable Equilibrium State is a crucial component of our method. In this paper, we emphasize that a stable supernet is essential for deriving robust architectures; while a supernet undergoing significant fluctuations with the precipitous validation loss landscape, would lead to a dramatic performance drop when deriving the final architecture, as also highlighted in [1][2].
>
> It's also true that our EI metric evaluates the relative significance of operations by independently assessing their influence on supernet stability and does not account for the intricate dependencies between operations, just as we noted in the manuscript and highlighted in the "Limitations" section.
> We validate the generalization performance of our approach through comprehensive experiments across three different datasets and six diverse search spaces, demonstrating that our method remains effective despite these limitations.
>
> [1] CHEN X, HSIEH C J. Stabilizing Differentiable Architecture Search via Perturbation-based Regularization[J]. Cornell University - arXiv,Cornell University - arXiv, 2020.
>
> [2] ZELA A, ELSKEN T, SAIKIA T, et al. Understanding and Robustifying Differentiable Architecture Search[J]. International Conference on Learning Representations,International Conference on Learning Representations, 2020.
>
> **Response to weakness2:**
>
> We understand your concern regarding the broader applicability of our method. Differentiable NAS and evolutionary NAS methods represent distinct branches within the NAS domain, each with different optimization strategies and requirements. The central objective of our study is to address the ambiguities surrounding the actual role and impact of architecture parameters within the DARTS framework. By resolving these ambiguities, our work proposes techniques specifically tailored to improve DARTS method. As such, our contributions are intentionally focused on advancing the differentiable NAS research field, providing new insights that we hope will inspire the development of more sophisticated differentiable NAS methodologies.
>
> We recognize that the seamless application of many differentiable NAS techniques, including ours, to evolutionary algorithms is challenging due to fundamental differences in their design and optimization processes. However, we believe there is still potential for adapting the idea of our method to pruning-based approaches. This adaptation would involve initially identifying a stable supernet and subsequently evaluate and iteratively prune less influential operations. We agree that further investigation into this adaptation would be valuable and could open up interesting avenues for future research.
>
> **Response to Question1:**
>
> Thank you for this insightful and important question. We are currently conducting additional experiments to analyze the impact of these hyper-parameters on the stability and effectiveness of our method. The results of these ablation studies will be included in the revised manuscript under the ablation study section. We appreciate your patience as we complete this additional work.
>
> **Response to Question2:**
>
> We appreciate your question and the opportunity to discuss broader adaptations of our method. As mentioned earlier, the primary objective of our study is to resolve the ambiguities surrounding the actual role and impact of architecture parameters within the DARTS framework, and the Stable Equilibrium State and the Equilibrium Influential (EI) metric developed are tailored to differentiable NAS. Consequently, it's challenging to seamlessly apply our techniques to non-differentiable NAS methods, such as Zen-NAS [1] and SWAP-NAS [2]. Those methods often rely on zero-shot proxies, training-free evaluation metrics, or discrete search algorithms, which contrast with the gradient-based optimization used in differentiable NAS.
>
> However, potential adaptations may involve reconceptualizing our stability framework to identify a quasi-stable state within the search phase of these methods. This could be explored through proxies that mimic stability in a non-differentiable context, perhaps by using pre-training analytics or meta-learning strategies. While this is beyond the current scope of our work, we agree that extending the concept to non-differentiable or training-free methods could lead to promising research directions.
>
> [1] Ming Lin, Pichao Wang, Zhenhong Sun, Hesen Chen, Xiuyu Sun, Qi Qian, Hao Li, and Rong Jin. Zen-nas: A zero-shot NAS for high-performance image recognition. ICCV 2021.
>
> [2] Yameng Peng, Andy Song, Haytham. M. Fayek, Vic Ciesielski and Xiaojun Chang . SWAP-NAS: Sample-Wise Activation Patterns for Ultra-fast NAS. ICLR 2024.

---

> ### Author Response · Authors · 2024-11-22
> **We have completed additional experiments and uploaded the latest version of the manuscript**
>
> Thanks for your insightful and important question. According to your suggestions, we have completed additional experiments to analyze the impact of these hyper-parameters on the stability and effectiveness of our method. And we synchronize the results with you, as shown below:
>
> Table 1 The performance of different learning rate in NAS-Bench-201
>
> |Learning rate  |Acc.（%）on CIFAR10|  Acc.（%）on CIFAR100 |  Acc.（%）on ImageNet-16 |
> | :---: |  :---:|  :---: |  :---: |
> | 0.025 | 94.37 |  73.09 | 46.63 |
> | 3e-3 | 94.08|  72.01 | 45.62 |
> | 1e-4 | 94.02|  73.00 | 45.44 |
>
> Table 2 The performance of different batch size in NAS-Bench-201
>
> |Batch size  |Acc.（%）on CIFAR10|  Acc.（%）on CIFAR100 |  Acc.（%）on ImageNet-16 |
> | :---: |  :---:|  :---: |  :---: |
> | 64 | 94.37 |  73.09 | 46.63 |
> | 32 | 94.24|  72.76 | 46.23 |
> | 16 | 94.36|  73.51 | 46.34 |
>
> We conducted an ablation study by setting the learning rates to 0.025, 3e-3, and 1e-4, and the batch sizes to 64, 32, and 16, respectively.  The performance of different learning rates in NAS-Bench-201 is shown in Table 1, and the performance of different batch sizes in NAS-Bench-201 is also examined, as shown in  Table 2. We observe that, although there are slight variations in performance due to different hyper-parameters, our method consistently identifies architectures with superior performance. This highlights the generality and robustness of BOSE-NAS.
>
> Moreover, the results of these ablation studies have been included in the revised manuscript under the Ablation Study section, where they are highlighted for your convenient review. We sincerely appreciate your inquiry, If you have any further questions or encouraging feedback, please do not hesitate to contact us. We look forward to hearing from you.

---

> ### Comment · Reviewer_uGSJ · 2024-11-22
> **Thanks for the response**
>
> Thank authors for the response, I'd like to keep the my rating due to several reasons:
>
> &nbsp;
> &nbsp;
>
> 1. Research on differentiable NAS has been going on for several years since the advent of DARTS in 2018, previous work was trying to mitigate the optimisation gap from different perspectives. To this point, the proposed EI metric indeed provides a new angle to solve aforementioned issue. However, the DARTS search space is well-designed, even random search can achieve decent results on it [1, 2]. Most of work which conducted on the DARTS space can achieve performance around 96.xx% to 97.xx% on the CIFAR-10 and 75.xx% to 76.xx% on the ImageNet datasets, the results of BOSE-NAS are not sufficient evidences to support the superiority of the proposed method.
>
> &nbsp;
>
> 2. The emergence of Transformer and pre-trained large language/vision models has had a huge impact on the CNN and NAS fields. The reason that I was curious about the potential extension of EI metric to non-differentiable methods, as there are obvious limitations of differentiable NAS method, which are the flexibility and generalisation. For example, non-differentiable methods can be more easily extended to Transformer architecture search [3] than the differentiable ones. Most of the DARTS-like methods are still conducted on the DARTS space or DARTS-like search spaces, e.g., NAS-Bench-201, as they are more rely on a specifically designed differentiable search space. Proposing a new method that solving a widely studied issue of differentiable NAS method in the CNN-based search space, I'm personally feeling that is insufficient to promote the NAS research.
>
> &nbsp;
>
> Once again, I'm appreciate the efforts that authors did in this work and in the rebuttal.
>
> &nbsp;
> &nbsp;
> &nbsp;
>
> [1] Liam Li and Ameet Talwalkar. "Random search and reproducibility for neural architecture search." In Proceedings of The 35th Uncertainty in Artificial Intelligence Conference, 2020.\
> [2] Yu, Kaicheng, Christian Suito, Martin Jaggi, Claudiu-Cristian Musat, and Mathieu Salzmann. "Evaluating the search phase of neural architecture search." In Eighth International Conference on Learning Representations. 2020.\
> [3] Zhou, Qinqin, Kekai Sheng, Xiawu Zheng, Ke Li, Xing Sun, Yonghong Tian, Jie Chen, and Rongrong Ji. "Training-free transformer architecture search." In Proceedings of the IEEE/CVF Conference on Computer Vision and Pattern Recognition. 2022.

---

> > ### Author Response · Authors · 2024-11-22
> > **Response to your insightful questions**
> >
> > **Response to Reason 1:**
> >
> > First, we appreciate the reviewer’s acknowledgment of the novelty of our work. Indeed, research on DARTS has been ongoing for several years, with significant improvements in search efficiency and performance. However, challenges remain, particularly the ambiguity surrounding the role and impact of architecture parameters, which is a foundational issue in DARTS-based frameworks.
> >
> > While it is true that BOSE-NAS achieves performance comparable to state-of-the-art (SOTA) methods in several experiments, we would like to emphasize that the central goal of our research extends beyond empirical accuracy. Our primary objective is to address a critical but under explored issue in differentiable NAS—the ambiguity surrounding the role and impact of architecture parameters. our work not only improves the theoretical understanding of differentiable NAS but also provides a robust foundation for designing more effective NAS methodologies. These contributions complement empirical performance with deeper scientific insights and add significant value to the field.
> >
> > We hope this response clarifies our perspective and underscores the importance of our contributions beyond raw performance metrics.
> >
> > **Response to Reason 2:**
> >
> > We acknowledge the transformative impact of Transformers and pre-trained large models across many AI research areas, including NAS. However, there has been growing interest in applying differentiable NAS methods to Transformer or LLM-related domains. For instance:
> >
> > 1) DARTSformer [1] introduces a multi-split reversible network combined with DARTS to efficiently search for optimal Transformer architectures.
> >
> > 2) DARTS-CONFORMER [2, 3] uses DARTS to optimize the Transformer-based Conformer model for end-to-end Automatic Speech Recognition (ASR).
> >
> > 3) Differentiable Model Scaling (DMS) [4] applies differentiable methods to optimize network width and depth for large language models (LLMs).
> >
> > Notably, DARTSformer reported that compared to non-differentiable NAS methods, DARTS demonstrated superior performance on large-scale Evolved Transformers, achieving significant computational savings by reducing search costs by an order of magnitude.
> >
> > These examples demonstrate the adaptability of differentiable NAS to non-CNN-based search spaces, and to Transformers/LLMs domains. By addressing the foundational issues in differentiable NAS, we believe our work could also contribute to advancing these broader applications.
> >
> > To further address your concern, we are currently running additional experiments on RNN-based search space and hopefully we would provide your with the latest results as soon as possible.
> >
> > [1] Zhao, Y., Dong, L., Shen, Y., Zhang, Z., Wei, F., & Chen, W. (2021). Memory-Efficient Differentiable Transformer Architecture Search. Findings of the Association for Computational Linguistics: ACL-IJCNLP 2021. doi:10.18653/v1/2021.findings-acl.3722.
> >
> > [2] Shi, X., Zhou, P., Chen, W., & Xie, L. (2021). Darts-Conformer: Towards Efficient Gradient-Based Neural Architecture Search For End-to-End ASR. CoRR, abs/2104.02868. Retrieved from https://arxiv.org/abs/2104.028683.
> >
> > [3] Shi, X., Zhou, P., Chen, W., & Xie, L. (2021). Efficient Gradient-Based Neural Architecture Search For End-to-End ASR. In Companion Publication of the 2021 International Conference on Multimodal Interaction (pp. 91-96). Association for Computing Machinery. doi:10.1145/3461615.34911094.
> >
> > [4] Liu, K., Wang, R., Gao, J., & Chen, K. (2024). Differentiable Model Scaling using Differentiable Topk. Proceedings of the 41st International Conference on Machine Learning.

---

> > > ### Comment · Reviewer_uGSJ · 2024-11-22
> > >
> > > Thanks for your rapid reply. Since you are aware the existence of differentiable Transformer architecture search, I recommend the authors to verify the BOSE-NAS and EI metric on Transformer-based search space rather than RNN-based. This will be more convincing and draw more attention.

---

> ### Author Response · Authors · 2024-11-25
> **Response to your insightful recommendation**
>
> Thanks for this insightful recommendation. In response to your suggestion, we applied BOSE-NAS to optimize the fine-tuning process of ALBERT [1], a large pre-trained Transformer-based model.
>
> The results shown in Table 1 demonstrate that BOSE-NAS efficiently identifies the optimal architecture of adapter module, achieving higher accuracy with fewer fine-tuned parameters compared to full fine-tuning of ALBERT. These results further confirm the effectiveness and adaptability of BOSE-NAS across different search spaces, including Transformers. These findings will be included in the Appendix of the revised manuscript to enhance the scope and impact of our work.
>
> Table 1 Accuracy and the number of parameters for different fine-tuning methods on ALBERT backbone.
> |Fine-tuning methods  |Acc.（%）on QNLI| Finetuned Params|
> | :---: |  :---:|  :---: |
> | Full-finetuning | 86.27 | 11,683,584 |
> | Adapter | 86.49|  617,856 |
> | Adapter+BoseNAS | 87.01|  631,296 |
>
> While we value the importance of demonstrating the generalizability of BOSE-NAS, we would like to reiterate that the central focus of our study is to resolve critical ambiguities surrounding architecture parameters in DARTS frameworks. By addressing this foundational issue, our work provides new theoretical insights that we believe will inspire the community to build more advanced and versatile differentiable NAS methods, potentially extending beyond the scope of BOSE-NAS itself.
>
> We are grateful for your recommendation, which has helped us further strengthen our manuscript. We hope the additional Transformer-based experiments, along with our focus on addressing foundational challenges, meet your expectations and add value to the differentiable NAS community.
>
> **The Appendix will be revised as:**
>
> PERFORMANCE IN TRANSFORMER-BASED SEARCH SPACE
>
> To verify the generalization and robustness of BOSE-NAS, we applied it to optimize the fine-tuning process of ALBERT [1], a large pre-trained Transformer-based model. Fine-tuning large pre-trained models is critical for transfer learning in various scenarios. However, this approach often suffers from parameter inefficiency when addressing multiple downstream tasks, as each task requires a separate model. Adapter [2] modules offer a more efficient alternative, introducing a small number of trainable parameters for each task while preserving scalability. The architecture of the adapter significantly impacts both performance and parameter efficiency. Selecting the optimal architecture manually, however, is resource-intensive and often suboptimal.
>
> To address this, we utilize BOSE-NAS to automate the search for adapter architectures, balancing accuracy and computational efficiency. The search space is defined as {Identity Mapping, Self-Attention Layer, 1D Convolutional Layer (Conv1x1), Multi-Layer Perceptron (MLP)}
>
> The experimental results, summarized in Table 1, demonstrate that BOSE-NAS efficiently identified the optimal adapter architecture, achieving higher accuracy with fewer fine-tuned parameters compared to traditional full fine-tuning approaches. These findings highlight the effectiveness of BOSE-NAS in balancing performance and efficiency, making it a valuable tool for improving fine-tuning processes in Transformer-based models.
>
>
> [1] Lan, Z., Chen, M., Goodman, S., Gimpel, K., Sharma, P., & Soricut, R. (2019). ALBERT: A lite BERT for self-supervised learning of language representations. arXiv preprint arXiv:1909.11942.
>
> [2] Houlsby, N., Giurgiu, A., Jastrzebski, S., Morrone, B., de Laroussilhe, Q., Gesmundo, A., Attariyan, M., & Gelly, S. (2019). Parameter-efficient transfer learning for NLP. arXiv preprint arXiv:1902.00751.

---

### Official Review · Reviewer_ZL8P · 2024-10-27

**Soundness:** 4
**Presentation:** 3
**Contribution:** 3
**Rating:** 6
**Confidence:** 3

**Summary:**

This paper focuses on Differentiable Architecture Search (DARTS). They conduct theoretical analysis over DARTS and propose a concept called Stable Equilibrium State. Upon it, they propose an effective framework called BOSE-NAS to identify the optimal state during the searching procedure. Experiment results show that the proposed method shows competitive results over state-of-the-art methods.

**Strengths:**

1. I think this paper focuses on a very important problem. DARTS is a very crucial framework in NAS, but it has some well-known problems. It is very important to have some theoretical analysis on this framework.
2. This author provides large-scale theoretical analysis, focusing on very important aspects, such as the stability of bi-level optimization, the loss trajectory, etc. I think the analysis is insightful.
3. The proposed method can reduce the search costs.

**Weaknesses:**

1. I think the figures in this paper can be polished to be more clear (maybe in the camera ready version).
2. The accuracy of the proposed method is just comparable with sota, but not superior to sota. I think it is not a serious problem, but I just list it as one weakness.

**Questions:**

I think overall this paper is good. Currently I give 6 since I have not checked the proof very carefully. I am willing to raise the score to 8 if the proof is proved to be right by other reviewers.

---

> ### Author Response · Authors · 2024-11-15
> **Response to the weaknesses and questions**
>
> We appreciate your constructive feedback and the time taken to review our manuscript. Below, we address each of your points to ensure a comprehensive revision.
>
> **Response to Weakness1:**
>
> Thank you for noting the potential improvement in figure clarity. We have revised all figures to enhance their visual quality by increasing the resolution and refining the labels and legends for better readability. These updated figures will be included in the revised version, ensuring that they convey the data more effectively.
>
> **Response to Weakness2:**
>
> We appreciate your acknowledgment of this aspect of our work. While it is true that our method achieves performance comparable to existing state-of-the-art (SOTA) methods in some of our experiments, we would like to emphasize that the primary contribution of our research extends beyond empirical accuracy.
>
> The central objective of our study is to address and resolve the ambiguities surrounding the actual role and impact of architecture parameters within the DARTS framework. This focus is critical for enhancing the theoretical understanding and robustness of differentiable NAS methods. We believe that filling these gaps and proposing a more analytically grounded differentiable NAS approach contributes significant value to the field, complementing empirical findings with deeper scientific insights. This combination of practical results and theoretical advancement broadens the understanding and potential applications of differentiable NAS methodologies.
>
> We hope this response clarifies our perspective and underscores the importance of our contributions beyond raw performance metrics.
>
> **Response to Question1:**
>
> Thank you for your encouraging assessment of our work and for noting the importance of verifying the theoretical proof. To support a thorough review, we will include a comprehensive and detailed explanation of our theoretical proof in the Appendix of the revised manuscript.

---

> ### Author Response · Authors · 2024-11-22
> **We have uploaded the latest version of the manuscript.**
>
> Thanks again for your valuable time and suggestions. According to your advice,  we have appended a comprehensive and detailed explanation of our theoretical proof in the Appendix of the revised manuscript. In addition, we have polished the figure to make it clearer. All revisions have been updated in the latest version of the manuscript and are highlighted for your convenient review.
>
> We hope our responses have effectively addressed your concerns. Should you have any additional questions or further feedback, we would be happy to continue the discussion. Additionally, we kindly ask you to reconsider your rating in light of our responses. Based on your comments, we understand that you have a positive view of our work, and we believe the points you raised have been thoroughly addressed in our responses.
>
> Thank you again for your valuable input, and we look forward to further discussions.

---

> > ### Comment · Reviewer_ZL8P · 2024-11-23
> >
> > After reading other reviews and the rebuttal, I hold my score.

---

### Meta-Review · Area_Chair_r9GW · 2024-12-24

**Metareview:**

This paper focuses on Differentiable Architecture Search (DARTS) and proposes BOSE-NAS, a differentiable neural architecture search method based on a stable equilibrium state. The article is easy to read and includes rich theoretical analysis, with the proposed method showing advantages in terms of efficiency. However, the unanimous opinion of several reviewers is that the paper lacks broader comparisons and more solid experimental results. The Area Chair reviewed the paper and all discussions and believes that the technical contribution of the study does not meet the acceptance standards and still requires improvement.

**Additional Comments On Reviewer Discussion:**

The main concerns raised by the reviewers were the lack of broader comparisons and concerns about the existing performance. During the defense, the authors provided some explanations and added additional experiments, but the reviewers still maintained their stance. The Area Chair also believes that the issue has not been fully addressed.

---

### Decision · Program_Chairs · 2025-01-22

Reject